# Coexistence of ferroelectricity and antiferroelectricity in 2D van der Waals multiferroic

Yangliu Wu[1], Zhaozhuo Zeng[2], Haipeng Lu[1], Xiaocang Han[3], Chendi Yang[4], Nanshu Liu[5], Xiaoxu Zhao [3], Liang Qiao [6], Wei Ji [5] ✉, Renchao Che [4], Longjiang Deng [1] ✉, Peng Yan [2] ✉ & Bo Peng [1] ✉

Multiferroic materials have been intensively pursued to achieve the mutual control of electric and magnetic properties. The breakthrough progress in 2D magnets and ferroelectrics encourages the exploration of low-dimensional multiferroics, which holds the promise of understanding inscrutable magnetoelectric coupling and inventing advanced spintronic devices. However, confirming ferroelectricity with optical techniques is challenging in 2D materials, particularly in conjunction with antiferromagnetic orders in single- and few-layer multiferroics. Here, we report the discovery of 2D vdW multiferroic with out-of-plane ferroelectric polarization in trilayer $NiI_2$ device, as revealed by scanning reflective magnetic circular dichroism microscopy and ferroelectric hysteresis loops. The evolution between ferroelectric and antiferroelectric phases has been unambiguously observed. Moreover, the magnetoelectric interaction is directly probed by magnetic control of the multiferroic domain switching. This work opens up opportunities for exploring multiferroic orders and multiferroic physics at the limit of single or few atomic layers, and for creating advanced magnetoelectronic devices.

Multiferroic materials with the coexistence of ferroelectric and magnetic orders have been diligently sought after for a long time to achieve the mutual control of electric and magnetic properties for energy-efficient memory and logic devices[1–3]. However, the two contrasting order parameters tend to be mutually exclusive within a single-phase material[4]. Nondisplacive mechanisms introduce a paradigm for constructing multiferroics beyond the traditional limits of mutual obstruction between ferroelectric and magnetic orders[5,6]. To date, type I multiferroic $BiFeO_3$ is the only known room-temperature single-phase multiferroic material. Alternatively, helical magnetic orders

break inversion symmetry and simultaneously lead to electric orders[7,8], giving rise to type-II multiferroics. The quest for an elusive single-phase multiferroic remains an open challenge.

The emergence of 2D vdW magnets and ferroelectrics has opened avenues for exploring low-dimensional physics in magnetoelectric coupling[9,10]. Diverse isolated vdW ferromagnets[11–13] and ferroelectrics[14,15] have enabled tantalizing opportunities to create 2D vdW spintronic devices with unprecedented performance at the limit of single or few atomic layers. A few bulk crystals of transition-metal dihalides with a trigonal layered structure have been shown to exhibit

[1]National Engineering Research Center of Electromagnetic Radiation Control Materials and Key Laboratory of Multi Spectral Absorbing Materials and Structures of Ministry of Education, School of Electronic Science and Engineering, University of Electronic Science and Technology of China, Chengdu, China. [2]School of Physics and State Key Laboratory of Electronic Thin Films and Integrated Devices, University of Electronic Science and Technology of China, Chengdu, China. [3]School of Materials Science and Engineering, Peking University, Beijing, China. [4]Laboratory of Advanced Materials, Department of Materials Science, Collaborative Innovation Center of Chemistry for Energy Materials(iChEM), Fudan University, Shanghai, China. [5]Beijing Key Laboratory of Optoelectronic Functional Materials & Micro-Nano Devices, Department of Physics, Renmin University of China, Beijing, China. [6]School of Physics, University of Electronic Science and Technology of China, Chengdu, China. ✉e-mail: wji@ruc.edu.cn; denglj@uestc.edu.cn; yan@uestc.edu.cn; bo_peng@uestc.edu.cn

helical spin textures that break inversion symmetry and induce orthogonal ferroelectric polarization[16,17], but definitive multiferroicity remains elusive at the limit of a few atomic layers.

Recent research shows the possibility of discovering type-II bilayer and even monolayer $NiI_2$ multiferroics using optical measurements such as second-harmonic generation (SHG) and linear dichroism (LD)[18,19]. Our previous work has shown that all-optical characterizations are not sufficient to make a judgment on few- or single-layer multiferroics in the presence of non-collinear and antiferromagnetic orders[20]. The observed SHG and LD signals in few-layer $NiI_2$ originate from the magnetic order[20–22]. Theoretically, SHG can manifest in various magnetic materials through inversion symmetry breaking induced by magnetic ordering, without necessitating ferroelectric properties[23]. The prerequisite for ferroelectricity is the non-vanishing spontaneous electric polarization, which must be proven through reliable and direct electrical measurements, such as polarization-electric field (*P-E*) and current-electric field (*I-E*) hysteresis loops. To date, 2D vdW multiferroics have not yet been directly uncovered at the limit of a few atomic layers. Here, we report fascinating vdW multiferroics with the coexistence of ferroelectricity and antiferromagnetism in few-layer $NiI_2$, based on magneto-optical-electric joint measurements. In this 2D vdW multiferroic, an exotic magnetic-field control of the switching dynamics of ferroelectric domains has been observed.

## Results and discussion

Due to the high reactivity of $NiI_2$ flakes, the exfoliation and encapsulation of $NiI_2$ by graphene and hexagonal boron nitride (hBN) flakes were carried out in a glove box (Fig. 1a and Supplementary Fig. 1). The $NiI_2$ crystal shows a rhombohedral structure with a repeating stack of three (I-Ni-I) layers, where Ni and I ions form a triangular lattice in each layer (Fig. 1b). The rhombohedral stacking has been atomically identified (Fig. 1c). The atomic arrangements of the rhombohedral phase demonstrate signature hexagon-shaped periodic bright spots with equal contrast, validating the overlapping stacking of I and Ni atoms along the **c**-axis. The ADF-STEM and fast Fourier transform (FFT) show an interplanar spacing of 1.9 Å, consistent with the (110) lattice plane of the rhombohedral $NiI_2$ crystal. Circularly polarized Raman spectra in the parallel (σ+/σ+ and σ-/σ-) configurations show only two distinct peaks in the $NiI_2$ device at room temperature (Fig. 1d). The peak at

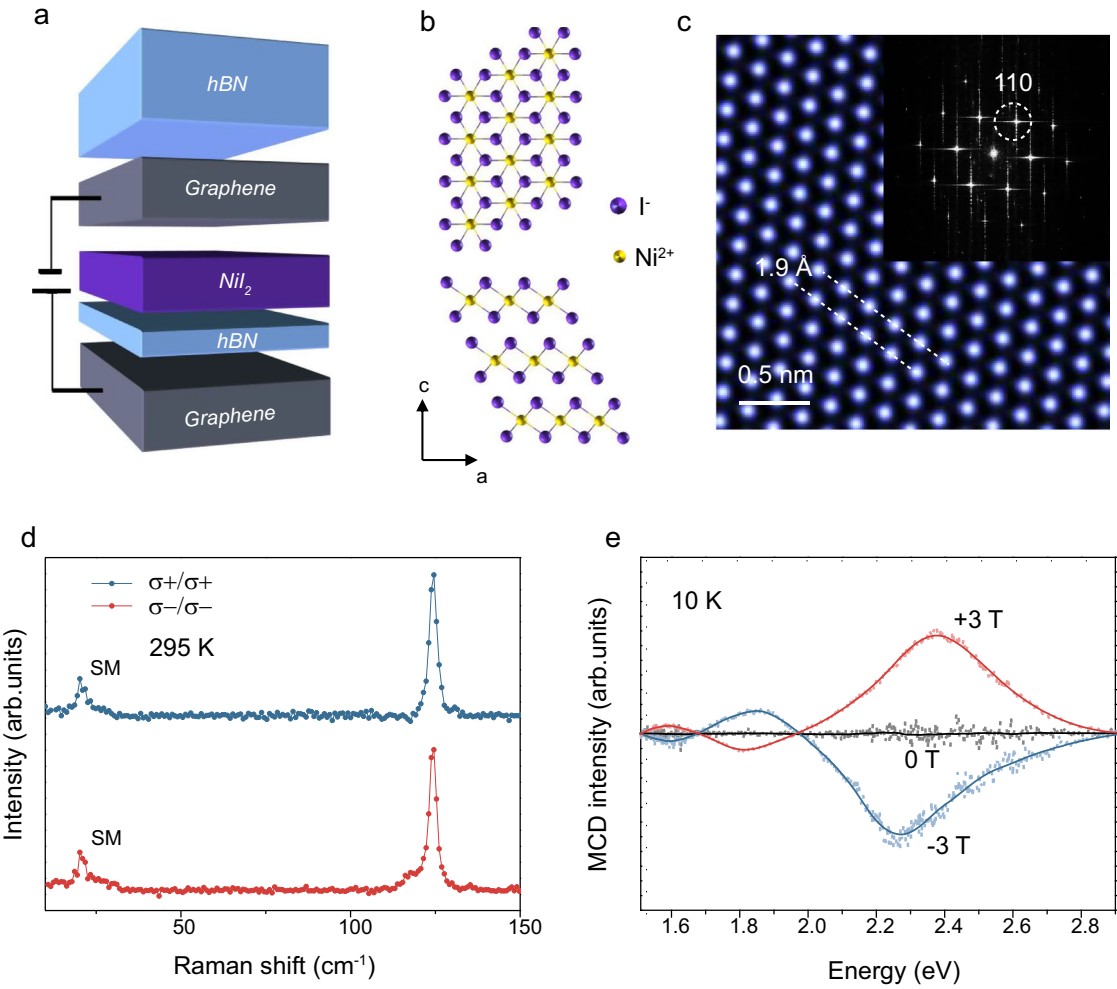

**Fig. 1 | Crystal structure, MCD measurements of trilayer $NiI_2$ at room temperature. a** Schematic of trilayer $NiI_2$ sandwiched between graphene and hBN. **b** View of the in-plane and out-of-plane atomic lattices. The magnetic $Ni^{2+}$ ions are surrounded by the octahedron of $I^-$ ions, and three $NiI_2$ layers as a repeating unit stack in a staggered fashion along the c-axis. **c** Atomic-resolution ADF-STEM image showing signature hexagonal patterns of rhombohedral stacking in few-layer $NiI_2$ crystals. The inset shows the corresponding FFT image. The white dashed lines highlight the interplanar spacing of 1.9 Å, while the white dashed circles emphasize the (110) lattice planes of the $NiI_2$ crystal. **d** Circular polarization resolved Raman spectra of a trilayer $NiI_2$ device (Fig. 1a) at room temperature, excited by 532 nm laser. "SM" indicates the interlayer shear mode of trilayer $NiI_2$. **e** The MCD spectra of trilayer $NiI_2$ at +3T, 0T and −3T. MCD signals are sensitive to spin electronic transitions and magnetic moments in the electronic states. The MCD features are spin-sign dependent and reverse as magnetic field switch. The zero remanent MCD signals at -2.3 eV at 0 T suggest antiferromagnetic orders. The dotted lines represent the raw data, while the solid lines indicate the smoothed data.

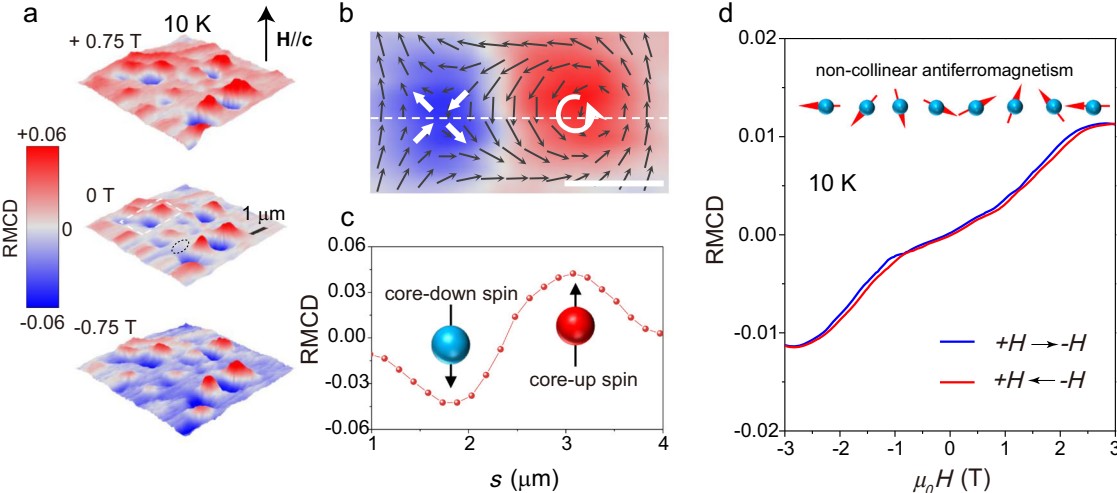

**Fig. 2 | Non-collinear antiferromagnetism in trilayer NiI₂ device. a** Polar RMCD maps upon a 2.33 eV laser with diffraction-limited spatial resolution (see Methods), collected at 10 K and selected magnetic field (the black arrows indicate the direction of the applied magnetic field). **b** Schematic of the spin textures of bimerons-like domains and corresponding zoom-in RMCD images (white dashed-line box in Fig. 2a). The black arrows represent the in-plane components, while colors denote the out-of-plane components (polarity). The helicity and chirality of the in-plane spin configurations are highlighted by white arrows. The scale bar is 1 μm. **c** The polar RMCD signals along with the line sections of RMCD map (**b**). The red and light blue spheres with black arrows indicate the spins of the core of bimeron-like domains. **d** The RMCD curves sweeping between −3 T and +3 T at 10 K, recorded in the area highlighted by dark dashed circle in Fig. 2a. The red (blue) curve corresponds to an increasing (decreasing) field. A schematic of the non-collinear antiferromagnet is shown in the inset.

-124.7 cm⁻¹ is assigned to the $A_{1g}$ phonon modes[24], and this polarization behavior is consistent with Raman tensor analysis for the rhombohedral structure of NiI₂ (Supplementary Fig. 2 and Supplementary Note 1)[25]. The Raman feature at ~20 cm⁻¹ is assigned to the interlayer shear mode (SM), which suggests that the NiI₂ is trilayer[21]. Furthermore, variable-temperature micro-Raman spectroscopy was performed in co-polarized channels (Supplementary Fig. 3a). A new peak at approximately 30 cm⁻¹, attributed to the magnon mode[24], was detected at 10 K. Additionally, the magnetic-order-induced magnon-phonon coupling leads to the emergence of the $E_g$ mode at around 78 cm⁻¹. With increasing temperature, the peaks at 78 and 30 cm⁻¹ gradually diminish and disappear around 35 K, suggesting a magnetic transition temperature of 35 K (Supplementary Fig. 3b). This observation is consistent with the magnetic transition temperature of trilayer NiI₂ reported in previous studies and further validates that the NiI₂ flakes have three layers[18,21,24].

For optimal optical response and sensitivity in probing the magnetic properties, the photon energy should be chosen near the absorption edge[11,26]. Therefore, we first studied the white-light magnetic circular dichroism (MCD) spectra of a trilayer NiI₂ device as a function of the magnetic field perpendicular to the sample plane at 10 K (see Methods for details)[27]. There is a strong peak near 2.3 eV along with two weak features around 1.85 eV and 1.6 eV (Fig. 1e). By means of ligand-field theory, the peaks are attributed to the absorption transitions of *p-d* exciton states[28]. A pair of opposite MCD peaks with the magnetic field clearly appears at 2.3 eV, suggesting a strong magneto-optical resonance. When the magnetic field is switched, MCD features are consistently reversed, and a zero remanent MCD signal at ~2.3 eV is distinctly observed at 0 T, indicating antiferromagnetic order at 10 K.

To further validate the magnetic order, the scanning RMCD microscope was used to image and measure the magnetic domains of the as-exfoliated trilayer NiI₂. Polar RMCD imaging is a reliable and powerful tool for unveiling 2D magnetism on the microscale, and the RMCD intensity is proportional to the out-of-plane magnetization[11,26]. All magneto-optical measurements were carried out using a 2.33 eV laser with optimal detection sensitivity (see Methods for details), and the applied magnetic field was oriented parallel to the **c**-axis. Figure 2a

shows RMCD maps of a trilayer NiI₂, sweeping between -0.75 T and +0.75 T at 10 K. Remarkably, many micrometer-sized bimeron-like domains are observed in the trilayer and other few-layer NiI₂ across the entire range of the sweeping magnetic field[29]. Spin-up and spin-down domains exist in pairs (Fig. 2a and Supplementary Fig. 4). One typical bimeron-like domain in trilayer NiI₂ at 0 T and 10 K is shown in Fig. 2b. The RMCD signals in each bimeron-like domain display opposite signs and nearly equal intensities. The magnetic moments point upwards or downwards in the core region, gradually decrease away from the core, and approach zero near the perimeter (Fig. 2c). This magnetic moment distribution possibly indicates a pair of topological spin meron and antimeron with opposite chirality in a cycloid ground state[30,31]. The bimeron-like magnetization textures remain robust in all magnetic fields, indicating that the bimeron-like domains are stable. The high stability of the bimeron-like magnetic domains probably originates from topological protection, which also contributes to the preservation of magnetization even upon a reversal of the magnetic field at 0.75 T. However, further in-depth studies are needed to reveal the exact physical mechanism.

Figure 2d shows the RMCD loops of the trilayer NiI₂ sweeping between +3 T and -3 T at 10 K, collected from the area enclosed by the black circle in Fig. 2a (middle panel). The RMCD loops show highly nonlinear behavior with the magnetic field and plateau behavior for the out-of-plane magnetization. The RMCD intensity near 0 T is suppressed and approaches zero, suggesting vanishing remnant magnetization, which indicates a compensation of the out-of-plane magnetization and non-collinear AFM orders in the trilayer NiI₂[32]. Gradual increases in the RMCD signal are observed with increasing magnetic field between ±1.2 T and ±2.6 T, suggesting a spin-flop process. The spin-flop behavior of the magnetization curve implies that the interlayer antiferromagnetic coupling of the non-collinear spins is complicated. Similar magnetic hysteresis loops have been observed in another few-layer NiI₂ sample, which suggests non-collinear AFM orders (Supplementary Fig. 4b).

To determine ferroelectricity in a few-layer NiI₂ device, we performed frequency-dependent measurements of electric polarization via *I-E* and *P-E* hysteresis loops, which allow for an accurate estimation of the electric polarization. We fabricated two heterostructure devices

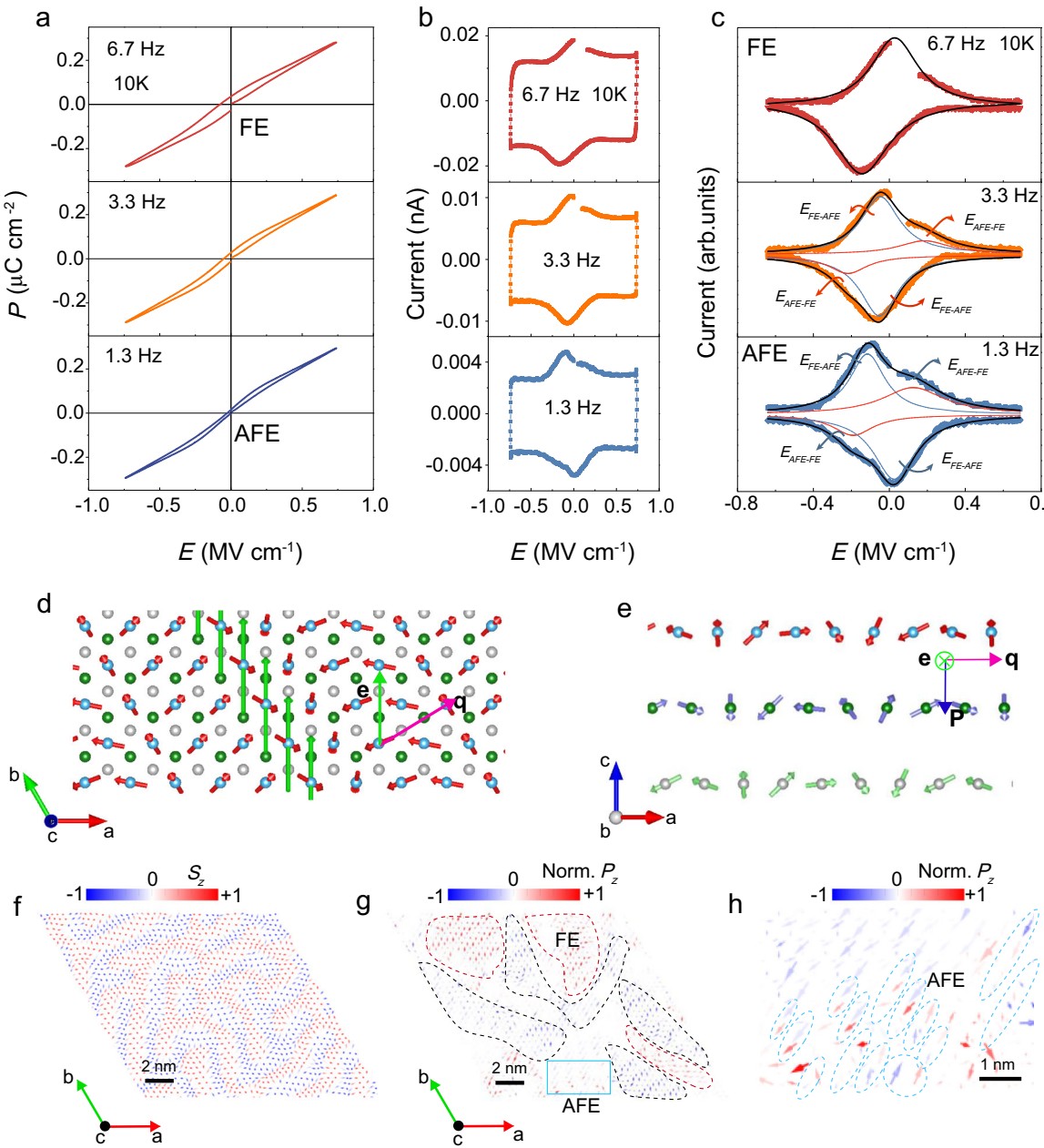

**Fig. 3 | Existence of ferroelectric and anti-ferroelectric orders in trilayer NiI$_2$ device. a, b** *P-E* and *I-E* loops at various frequencies from device 1. **c** Corresponding *I-E* loops from Fig. 3b subtracted the current background. Two pairs of current peaks (FE-AFE and AFE-FE switching peaks) were obtained by Lorentz fitting. **d** Schematics of spiral configuration in top views. The light blue spheres and red arrows represent the Cr atoms in the top layer and their spins. **e** Schematics of spiral configuration in side views, showing the interlayer AFM coupling and out-of-palne ferroelectricity. **f** Spin texture of NiI$_2$ from atomistic spin model simulations. Spins

are represented by arrows, with the red and blue colors showing positive and negative values of the $S_z$ component. **g** Corresponding ferroelectric dipole textures obtained through numerical computations. Dipoles are represented by arrows, with the red and blue colors showing positive and negative values of the $P_z$ component. The areas encircled by red and black dashed lines highlight FE domains with downward and upward $P_z$ components, respectively. **h** An enlarged view of the out-of-plane AFE domain occurring in panel **g** (light blue solid line box). The light blue dashed circles highlight pairs of opposing $P_z$.

of graphene/hBN/NiI$_2$/graphene/hBN (Fig. 1a and Supplementary Fig. 1). The hBN flake was used as an excellent insulating layer to prevent large leakage currents and to ensure the detection of ferroelectric (FE) features[33,34] (Supplementary Fig. 5 and 6; Supplementary Note 2). The hBN insulator shows linear *P-E* behavior and rectangular-shaped *I-E* loops (Supplementary Fig. 7), indicating excellent insulating properties for ferroelectric hysteresis measurements (see Methods for details)[35,36]. The frequency-dependent *I-E* and *P-E* loops at 10 K are shown in Fig. 3, and the forward and backward scans of the electric polarization as a function of electric field show characteristic

ferroelectric *I-E* and *P-E* hysteresis. Strikingly, a double hysteresis loop behavior, which is a typical characteristic of an AFE[37,38], was observed in trilayer NiI$_2$ as the measured frequencies decreased. More importantly, a pair of opposite single peaks in switching current (*I*) is observed when sweeping voltage at 6.7 Hz, which is attributed to charge displacement and implies two stable FE states with inverse polarity (Fig. 3b, c). Whereas a pair of opposite bimodal peaks is observed when sweeping voltage at 1.3 Hz, which is attributed to AFE-FE and FE-AFE transitions[37] under electric field sweeping (Fig. 3c). This suggests that an evolution from FE to AFE with frequency is observed[38,39], providing the decisive

evidence for coexistence of ferroelectric and antiferroelectric states[40,41]. These comprehensive frequency-dependent evolution behaviors are also observed in another few-layer and bulk-like $NiI_2$ (Supplementary Figs. 8 and 9). These results confirm the coexistence of FE and AFE.

Type-II multiferroicity has been demonstrated in bulk $NiI_2$[16]. However, the multiferroic identification for few-layer $NiI_2$ remains challenging and elusive. All-optical methods are unreliable for making a judgment of a few- or single-layer multiferroic in the presence of non-collinear and antiferromagnetic orders[20]. The bulk $NiI_2$ displays a helimagnetic state with spin propagation $\mathbf{q} = (0.138, 0, 1.457)$ in a transformation coordinate below critical temperature[42,43] and actually the projection of the $\mathbf{q}$-vector in $\mathbf{ab}$-plane makes an angle of 30° with the $\mathbf{a}$-axis along the [210] direction. From symmetry considerations and a Ginzburg-Landau perspective[44,45], the helimagnetic state allows for the emergence of a ferroelectric polarization associated to the form:

$$\mathbf{P} = \gamma \mathbf{e} \times \mathbf{q} \tag{1}$$

where $\mathbf{P}$ is the electric polarization, $\mathbf{e}$ is the spin rotation axis, $\mathbf{q}$ is the spin propagation vector of the spin spiral, and $\gamma$ is a scalar parameter dependence with spin-orbit coupling. However, the magnetic ground state of few-layer $NiI_2$ remains elusive[46–49]. The $\mathbf{q}$-vector in multi-layer and bulk $NiI_2$ is primarily determined by the competition among various magnetic exchange interactions between magnetic atoms[47,50]. In monolayer $NiI_2$, there are no interlayer interactions; thus, the $\mathbf{q}$-vector lies in the $\mathbf{ab}$-plane and is dictated by the intralayer exchange interactions[46]. The magnetic ground state of trilayer $NiI_2$ was investigated using density functional theory (DFT) calculations (see supplementary text for details), revealing a spiral spin configuration with an $\mathbf{ab}$ in-plane projection of the $\mathbf{q}$-vector along the [210] direction (pink arrow in Fig. 3d). For trilayer $NiI_2$, which has already reached the bulk limit, the in-plane projection of the $\mathbf{q}$-vector is akin to that of the bulk material. To show a clearer view of the magnetic orders, only the top Ni layer is illustrated in Fig. 3d, and the magnetic moments of the trilayer Ni are presented in Supplementary Fig. 10. The interlayer AFM coupling in trilayer $NiI_2$ is attributed to the larger second interlayer nearest-neighbor exchange parameter (Fig. 3e). The orientation of the spin-rotation plane is predominantly determined by the Kitaev interaction[51,52], and the in-plane projection of the $\mathbf{e}$-vector is perpendicular to the direction of $\mathbf{a}$-axis (green arrow in Fig. 3d). The geometric arrangement of $\mathbf{q}$ and $\mathbf{e}$ results in an out-of-plane component ($\mathbf{c}$-axis direction) of ferroelectric polarization (Fig. 3d, e), consistent with the experimentally observed out-of-plane ferroelectricity (Fig. 3a).

Atomistic spin model simulations have been performed on large supercells using the effective field (Supplementary Text: Atomistic spin model simulations) and revealed that spiral spin states give rise to stripy domains, further indicating that the spin spirals are the ground states of trilayer $NiI_2$ (Fig. 3f). This result is analogous to the stripe domains induced by spiral spin states in monolayer $NiI_2$ as predicted by Monte Carlo simulations[52]. To clearly show the magnetic domain structures, the spin textures in each layer of the trilayer $NiI_2$ are presented in Fig. 3f and Supplementary Fig. 11a and 11c, respectively. The ferroelectric dipole orderings corresponding to the spiral spin textures can be calculated according to the Katsura-Nagaosa-Balatsky (KNB) or inverse Dzyaloshinskii-Moriya (D-M) mechanism[53,54]. Numerical simulations have demonstrated the coexistence of ferroelectric domains with opposite $P_z$ components, attributed to the distinct chirality of the magnetic domains (the areas enclosed by the red and black dashed lines in Fig. 3g), because the spiral magnetic states with opposite chirality are degenerated, but create opposite ferroelectric polarization[46,51]. In certain transitional regions, an intriguing phenomenon occurs where local dipoles with opposing $P_z$ components are interlaced, indicating the origin of antiferroelectricity (Fig. 3h). The calculated intermingling of ferroelectric and antiferroelectric domains supports the experimentally observed coexistence behaviors of ferroelectricity and antiferroelectricity. The distinct switching dynamics of different domain types likely account for the pronounced evolution between ferroelectric and antiferroelectric behaviors with frequency. The interlayer AFM coupling induces opposite magnetic moment components in adjacent layers, but the spin chirality is maintained in each individual layer (Fig. 3e). Therefore, the spatial distributions of the $S_z$ components in adjacent layers exhibit opposite characteristics, yet the polarization dipole textures remain predominantly unaltered (Fig. 3f, g, and Supplementary Fig. 11).

To reveal the magnetoelectric coupling effect, we studied the magnetic control of ferroelectric properties in the trilayer $NiI_2$ device, as shown in Fig. 4. The remanent polarization ($P_r$) extracted from the $P$-$E$ hysteresis loop is plotted as a function of the out-of-plane magnetic field at different frequencies (Fig. 4a–c). The out-of-plane magnetic field causes a decrease in $P_r$ at various frequencies (Fig. 4a–c and Supplementary Fig. 12), and the magnetic control of $P_r$ shows frequency dependence on the applied electric field (Fig. 4d). The magnetic control ratio reaches ~7% by tuning the frequency (24.5 Hz) at 7 T, showing a typical feature of type II multiferroics. To better understand the magnetic control behavior, we briefly discuss the possible mechanism from a microscopic perspective, focusing on ferroelectric polarization flops in the model of spiral magnets[44]. As illustrated in Supplementary Fig. 13, the spins exhibit rotation in the $\mathbf{ac}$-plane ($\mathbf{e}$ // $\mathbf{b}$), forming a transverse spiral with $\mathbf{q}$ // $\mathbf{a}$, thereby realizing an out-of-plane ferroelectric polarization ($\mathbf{P}$ // $\mathbf{c}$, top panel). When a magnetic field ($\mathbf{H}$ // $\mathbf{c}$) is introduced, the magnetization ($\mathbf{M}$ // $\mathbf{H}$ // $\mathbf{c}$) is induced, resulting in a transverse conical spin configuration with an effective $ab$ rotation plane where $\mathbf{e}$ // $\mathbf{H}$ // $\mathbf{c}$ (middle panel). Consequently, a reorientation of $\mathbf{P}$ from $\mathbf{P}$ // $\mathbf{c}$ to $\mathbf{P}$ // $\mathbf{b}$ takes place[44,55], and the out-of-plane ferroelectric polarization decreases. This scenario is consistent with the experimental observations that an out-of-plane magnetic field in trilayer $NiI_2$ leads to an increase in out-of-plane magnetization (Fig. 2d) and a decrease in out-of-plane ferroelectric polarization (Fig. 4a–c). Moreover, a decrease in the current peak (Fig. 4e and Supplementary Fig. 14), along with an increasing coercive electric field with the increasing magnetic field, is clearly evident (the background current is attributed to displacement current; Supplementary Note 3). This is because the out-of-plane magnetic field causes the $\mathbf{e}$-vector to tilt towards the $\mathbf{c}$-direction, reorienting the easy axis of ferroelectric polarization from the out-of-plane to the in-plane direction. The switching time of the ferroelectric domain under varying magnetic fields at 10 K is calculated using the KAI model (Fig. 4f, g). The switching time $\tau$ increases with the magnetic field, indicating an even symmetry with respect to the magnetic field (Fig. 4h), which is in line with the aforementioned mechanisms. At 10 K, the switching time $\tau$ results in a maximum enhancement of 20% (-7 T). When the $\mathbf{H}$ is applied parallel to the $\mathbf{q}$, a longitudinal conical spin configuration with an effective spin-rotation plane in the $\mathbf{bc}$-plane is established (bottom panel in Supplementary Fig. 13). In the scenario where $\mathbf{e}$-vector is parallel to $\mathbf{q}$ ($\mathbf{e}$ // $\mathbf{q}$), the ferroelectric polarization is anticipated to be zero according to the inverse D-M model ($\mathbf{P} \propto \mathbf{e} \times \mathbf{q}$). To further study the magnetoelectric coupling mechanism, in-plane magnetic fields have been applied to manipulate the ferroelectric polarization, which employs the parallel and perpendicular to the $S$-direction (sample orientation), as illustrated in Supplementary Fig. 15a and c. Interestingly, for $\mathbf{H}$ // $S$-direction, the out-of-plane ferroelectric polarization decreases with increasing magnetic field (Supplementary Fig. 15b). In stark contrast, for $\mathbf{H} \perp S$-direction, the ferroelectric polarization increases as the magnetic field increases (Supplementary Fig. 15d). The DFT calculations have revealed that the magnetic ground state is a spiral configuration with a propagation vector $\mathbf{q}$ along the [210]

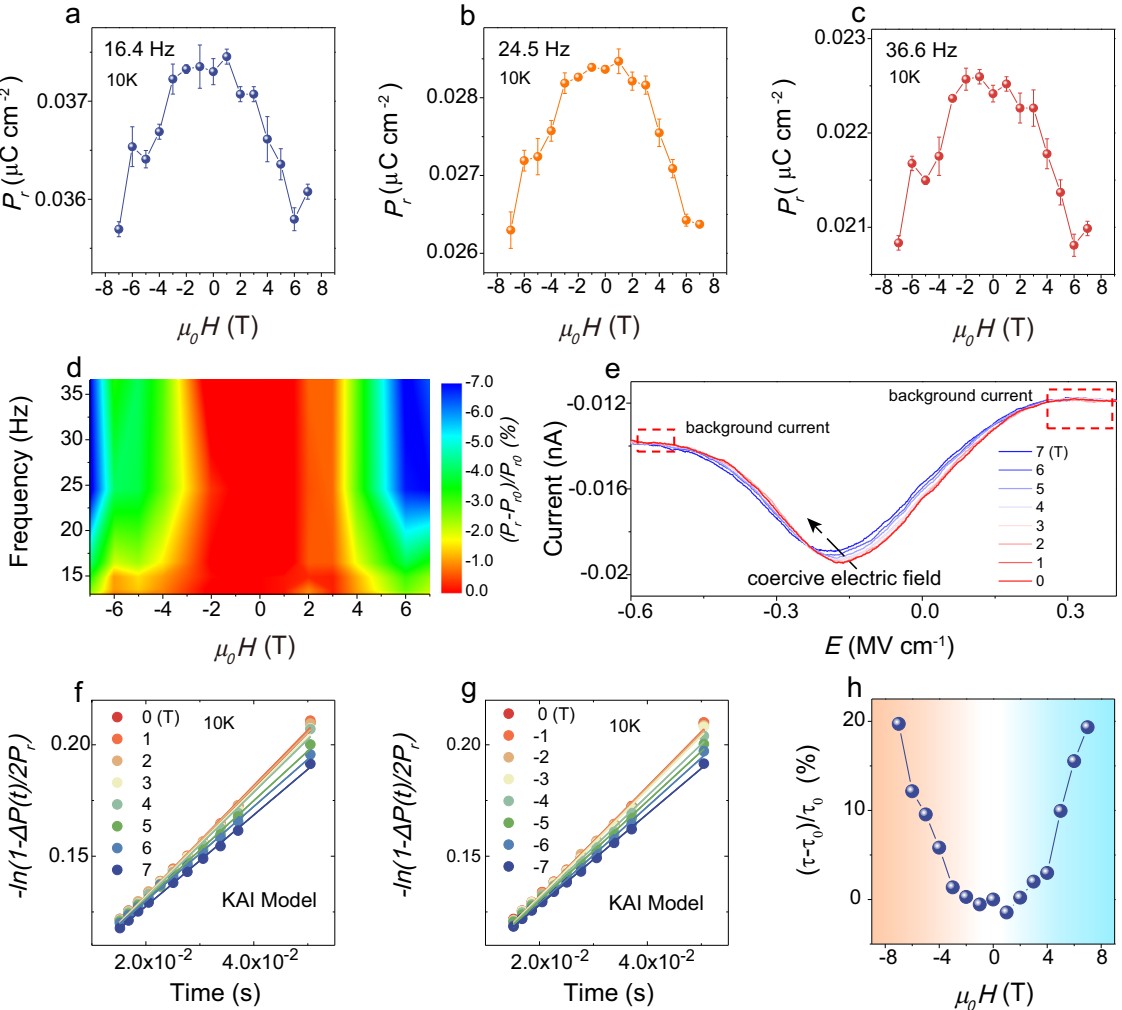

**Fig. 4 | Magnetic control of ferroelectricity in trilayer NiI₂ device. a–c** $P_r$ as a function of the out-of-plane magnetic field at different frequencies. For all relevant panels error bars represent mean ± standard error of the mean. **d** The phase diagram of the magnetic control ratio, $(P_r - P_{rO})/P_{rO}$, by tuning both the frequency and magnetic field, where $P_r$ and $P_{rO}$ represent the remanent polarization with and without the magnetic field, respectively. **e** I-E curves at different magnetic field. The black dashed arrows indicate the shift in current peaks with increasing magnetic field, while the red dashed boxes highlight the background current that remains unchanged as the magnetic field increases. **f, g** Theoretical fitting with the KAI model for different magnetic fields at 10 K. **h** $(\tau - \tau_0)/\tau_0$ as a function of the magnetic field at 10 K, where $\tau$ and $\tau_0$ represent the switching time with and without the magnetic field, respectively.

direction (indicated by the pink arrow in Supplementary Fig. 15e, f), while the **e**-vector is perpendicular to the **a**-axis, denoted by the green arrow. When an in-plane magnetic field is oriented at an azimuth between the **e**-vector and the **q**-vector, it steers the **e**-vector to align parallel to the magnetic field direction, leading to a decrease in the angle between the **e**-vector and the **q**-vector, thus diminishing the ferroelectric polarization (Supplementary Fig. 15g). Conversely, when a magnetic field is perpendicular to the $S$-direction, it causes an increase in the angle between the **e**-vector and the **q**-vector, resulting in an enhancement of the ferroelectric polarization (Supplementary Fig. 15h). This manipulation of ferroelectric properties by a magnetic field highlights the potential of utilizing few-layer NiI₂ as a platform for investigating magnetoelectric coupling physics in the two-dimensional limit and for developing advanced nano-magnetoelectric devices[56].

In summary, we report a 2D vdW single-phase multiferroic NiI₂ few-layer crystal. We observed strong evidence for the coexistence of ferroelectric and non-collinear antiferromagnetic order via RMCD, P-E, and I-E hysteresis loops. We achieve exotic magnetic control of ferroelectric properties in the NiI₂ trilayer. We envision that the 2D vdW single-phase multiferroic NiI₂ will provide numerous opportunities for

exploring fundamental low-dimensional physics and for creating ultra-compact magnetoelectric devices.

## Methods

### Sample fabrication

NiI₂ flakes were mechanically exfoliated from bulk crystals via PDMS films in a glovebox, which was synthesized by chemical vapor transport method from elemental precursors with molar ratio Ni:I = 1:2. All exfoliated hBN, NiI₂, and graphene flakes were transferred onto pre-patterned Au electrodes on SiO₂/Si substrates one by one to create heterostructure in the glovebox, which were further in-situ loaded into a microscopy optical cryostat for magneto-optical-electric joint-measurement. The whole process of NiI₂ sample fabrications and magneto-optical-electric measurements was kept out of atmosphere.

### Magneto-optical-electric joint-measurement

The polar RMCD, white-light MCD, Raman measurements, and ferroelectric P-E and I-E measurements were performed on a powerful magneto-optical-electric joint-measurement scanning imaging system (MOEJSI)[20], with a spatial resolution reaching diffraction limit. The MOEJSI system was built based on a Witec Alpha 300 R Plus

low-wavenumber confocal Raman microscope, integrated with a closed cycle superconducting magnet (7T) with a room temperature bore and a closed cycle cryogen-free microscopy optical cryostat (10 K) with a specially designed snout sample mount and electronic transport measurement assemblies.

The Raman signals were recorded by the Witec Alpha 300 R Plus low-wavenumber confocal Raman microscope system, including a spectrometer (150, 600, and 1800/mm) and a TE-cooling Andor CCD. A 532 nm laser of ~0.2 mW is parallel to the X-axis (0°) and focused onto samples by a long working distance 50× objective (NA = 0.55, Zeiss) after passing through a quarter-wave plate (1/4λ). The circular polarization-resolved Raman signals passed through the same 1/4λ waveplate and a linear polarizer, obtained by the spectrometer (1800/mm) and the CCD.

For white-light MCD measurements, white light with Köhler illumination from Witec Alpha 300 R Plus microscope was linearly polarized at 0o by a visible wire grid polarizer, passed through an achromatic quarter-wave (1/4λ) plate, and focused onto samples by a long working distance 50× objective (Zeiss, NA = 0.55). The right-handed and left-handed circularly polarized white light was obtained by rotating 1/4λ waveplate at +45° and -45°. The white-light spectra were recorded by the Witec Alpha 300 R Plus confocal Raman microscope system (spectrometer, 150/mm). The absorption spectrum of $NiI_2$ was obtained by normalizing the sample spectrum to that of the bare substrate[27]. By measuring the difference in the absorption spectra of right-handed and left-handed circularly polarized light under different magnetic fields, the corresponding magnetic circular dichroism (MCD) spectra can be obtained.

For polar RMCD measurements, a free-space 532 nm laser (2.33 eV) of ~2 μW modulated by photoelastic modulator (PEM, 50 KHz) was reflected by a non-polarizing beamsplitter cube (R/T = 30/70) and then directly focused onto samples by a long working distance 50× objective (NA = 0.55, Zeiss), with a diffraction limit spatial resolution of ~590 nm. The reflected beam which was collected by the same objective passed through the same non-polarizing beamsplitter cube and was detected by a photomultiplier (PMT), which was coupled with a lock-in amplifier, Witec scanning imaging system, superconducting magnet, voltage source meter, and ferroelectric tester.

Ferroelectric P-E and I-E hysteresis loop of a $NiI_2$ device of Gr/hBN/$NiI_2$/Gr were measured by classical ferroelectric measurements and directly recorded by ferroelectric tester (Precision Premier II: Hysteresis measurement), which were contacted with the top and bottom graphene electrodes by patterned Au electrodes (Fig. 1a) through the electronic assemblies of the microscopy optical cryostat. The mechanism of ferroelectric measurement has been given by previous work[57]. The detected signals include two components: a ferroelectric term of $NiI_2$ ($2P_rA$) and a linear non-ferroelectric term of hBN insulator (σEAt), Q = QNiI + QBN = $2P_rA$ + σEAt. If only hBN insulator, a linear P-E loop take place, consistent with our experimental results of hBN flake (Supplementary Fig. 7). The linear hBN background have no effect on the ferroelectric features, and hBN flakes as excellent insulator suppress and overcome the leakage current, which for guarantee the detections of $NiI_2$ ferroelectric features[33–36].

## STEM Imaging, Processing, and Simulation

Atomic-resolution ADF-STEM imaging was performed on an aberration-corrected JEOL ARM 200 F microscope equipped with a cold field-emission gun operating at 80 kV. The convergence semiangle of the probe was around 30 mrad. Image simulations were performed with the Prismatic package, assuming an aberration-free probe with a probe size of approximately 1 Å. The convergence semiangle and accelerating voltage were in line with the experiments. The collection angle for ADF imaging was between 81 and 228 mrad. ADF-STEM images were filtered by Gaussian filters, and the positions of atomic columns were located by finding the local maxima of the filtered series.

## Reporting summary

Further information on research design is available in the Nature Portfolio Reporting Summary linked to this article.

## Data availability

The data that support the findings of this study are available within the paper and Supplementary Information and have been deposited in a Figshare online repository. Any further data and materials required to reproduce the work are available from the corresponding authors upon reasonable request. Source data are provided with this paper.

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

## Acknowledgements

B.P. and L.D. acknowledge support from the National Science Foundation of China (52021001). B.P. acknowledges support from the National Science Foundation of China (62250073, 62450003). R.C.C. acknowledges support from the National Science Foundation of China (52231007). H.L. acknowledges support from the National Science Foundation of China (51972046). L.D. acknowledges support from the Sichuan Provincial Science and Technology Department (Grant no. 99203070). L.Q. acknowledges support from the National Science Foundation of China (520720591 and 11774044). J.W. thanks the National Natural Science Foundation of China (Grant no. 11974422), the Strategic Priority Research Program of the Chinese Academy of Sciences (Grant no. XDB30000000). P.Y. was supported by the National Key R&D Program under Contract No. 2022YFA1402802 and the National Natural Science Foundation of China (NSFC) (Grants nos. 12374103, 12434003, and 12074057).

## Author contributions

B.P. conceived the project. Y.W. prepared the samples and performed the magneto-optical-electric joint-measurements and Raman measurements assisted by B.P., and performed the ferroelectric measurements assisted by L.Q., and analyzed and interpreted the results assisted by H.L., N. L., W.J., L.D., P.Y and B.P. Z.Z., N.L., W.J. and P.Y. performed the theory calculations. C.Y, R.C, X.Z. and X.H. performed the STEM measurements. Y.W. and B.P. wrote the paper with input from all authors. All authors discussed the results.

## Competing interests

The authors declare no competing interests.
