## [Peer Review file · Nature Communications]

Coexistence of ferroelectricity and antiferroelectricity in 2D van der Waals multiferroic

Corresponding Author: Professor Bo Peng

Version 0:

Reviewer comments:

Reviewer #1

(Remarks to the Author)

In the manuscript entitled "Coexistence of ferroelectricity and antiferroelectricity in 2D van der Waals multiferroic" by Y. L. Wu et al., the authors reported the direct electrical measurements of spontaneous polarization in 2D vdW NiI₂ trilayers. By analysing the P-E and I-E hysteresis loops, the author concluded that both ferroelectric and antiferroelectric phases coexisted in the NiI₂ trilayers. And the out-of-plane ferroelectric polarization can be tuned by an out-of-plane magnetic field. The result and the interpretation look interesting to me. However, I notice that the multiferroicity order in NiI₂ monolayer or few layers has been extensively studied in recent years (10.1038/s41586-021-04337-x, 10.1088/2053-1583/ac4e9d, 10.1002/adma.202311342, 10.1002/adma.202109144, 10.1021/acs.nanolett.1c01095, 10.1021/acs.nanolett.1c01095, 10.1103/PhysRevLett.131.036701). Though the statement of antiferroelectric polarization would be a new finding, it lacks sufficient experimental or theoretical support in the present manuscript. Therefore, the novelty of this manuscript should be further clarified. And there are also some questions that the authors need to address:

1. The Raman spectra of the NiI₂ device only show two peaks at 124.7 cm⁻¹ and 20 cm⁻¹. At what temperature does the author perform the Raman spectra? Why the Raman peak of Eg symmetry around 75~80 cm⁻¹ (as reported in previous works 10.1038/s41586-021-04337-x, 10.1021/acsnano.0c04499) can not be identified in the Raman spectra? I also suggest the author to measure the temperature-dependent Raman spectra for their NiI₂ trilayers. Whether the Raman response in the magnetically ordered phase can be seen at low temperatures. This is important for confirming the AFM order in their NiI₂ devices.
2. The direction of applied magnetic field for the RMCD imaging should be clearly pointed out in the figures. The RMCD sweeping loops in Fig. 2d and Fig. S2b are obtained from the average value of whole mapping area or just from a local area? If the latter, the special area should be marked in the mapping image.
3. The author said that "The RMCD intensity near 0 T is suppressed and approaches zero, suggesting the vanishing remnant magnetization, which indicates a compensation of the out-of-plane magnetization and non-collinear AFM orders in the trilayer NiI₂". I think this can not be sufficient evidence for the AFM order in the NiI₂ trilayers. It could be also due to the strong in-plane anisotropy, which becomes dominant at low magnetic field. So more evidence for the AFM order is needed.
4. In Fig. 3, the author claimed only a single peak were observed when sweeping voltage at 6.7 Hz. This is not true. The broad shoulder peak around +0.1 MVcm⁻¹ for the I-E curve at 6.7 Hz is very similar to the I-E curves at 3.3 Hz. Moreover, why the FE-AFE transition is more significant at low frequency needs further explanation.
5. Though the author concluded the direct electric measurement was more reliable than the general SHG and LD measurement, I think they should also measure the SHG and LD signals for their NiI₂ devices. This comparison between SHE, LD and to P-E/I-E loops may give us some hints on the advantages of direct electric measurement.
6. The author claimed that the easy axis would tilt from the x-y plane of the NiI₂ monolayer to the x-z plane of the NiI₂ trilayer, and an out-of-plane magnetic field can rotate the spin back to x-y plane. However, these are all hypothesis, lacking of either experimental evidence of theoretical calculations.
7. If the magnetic field is applied in-plane, what will happen to the ferroelectric properties of NiI₂ device. According to the hypothesis of spin easy plane, the in-plane field would not change the P?
8. The antiferroelectric phase is interesting finding. Can the author give more provement?
9. I don't understand the conclusion "The shifts of current peaks induced by ferroelectric switching vary with the magnetic field, but the background current remains constant, excluding the magnetoresistance effects". In my opinion, the resistance

of NiI2

layer should be different at zero field and at a very high field of 7 T. So the different background current is the normal case. Please give the magnetoresistance vs. H curves for the NiI2 device at both 10 K and 300 K, to confirm the unchanged magnetoresistance below TN.

10. Fig. S2c-e show the RMCD maps for a few-layer NiI2. Except for the bimerons-like domains, the RMCD signal in other area is inconsistent with the RMCD sweeping in Fig. S2b. For example, positive signal is observed at 0 T in d, and the zero signal is observed at -0.75 T in e.

11. The author used the hBN flake as the insulating layer to prevent large leakage current. However, the I-E curves given in Fig. S4c indicated that the leakage current of hBN layer is about 1~4 nA. While for the hBN/NiI2 device in Fig. 3b, the leakage current is only ~0.01 nA. It implies that the NiI2 trilayer is more resistive than the hBN layer. So I doubt the necessity of hBN inserting. Have the authors measure the P-E and I-E curves for the Gr/NiI2/Gr devices?

For these reasons, I think this manuscript in current version does not merit publication in Nature Communications.

Reviewer #2

(Remarks to the Author)

Wu et al. investigated a few layer NiI2 flakes, and demonstrated a coexistence of ferroelectricity and antiferroelectricity as well as a magnetic control of the ferroelectricity. Multiferroicity in 2D limit is an interesting and important issue, and authors have discussed it using an appropriate way. Followings are my comments about several issues to be clarified.

Authors concluded the antiferromagnetic orders at 10 K from the MCD result showing zero remnant signal. This result can deny the ferromagnetic order, but does this already exclude any possibility to have a paramagnetic or diamagnetic state instead of the antiferromagnetic state?

Authors stated that magnetic moments point upwards or downwards in the core region. What is the evidence of this argument?

What is the evidence of the non-collinear AFM order in the trilayer NiI2?

Can authors have any information about the q-vector in the magnetic ordered state? This information is crucial in discussing the ferroelectricity based on Eq. (1).

What is the length scale of Fig. 2b? And, on which area was the result of Fig. 2d obtained?

I-E and P-E curves seem clear. To strengthen the arguments of authors, there should be another measurement results of (bulk-like) thicker flakes which should exhibit typical behaviors of FE order with an absence of AFE order.

The arguments about the antiferroelectric state is not clear. Authors seem to explain simply double-well potential in the FE state and a possible spatial coexistence of negative and positive polarizations. As this explanation can be applied to the multi-domain state in the ferroelectric state, I would like to see an additional explanation for the antiferroelectric state.

Authors observed 7% change of the polarization using a magnetic field, and they explained this observation based on a tilt of the spin rotation plane and the corresponding polarization flop. Does this process correspond to the polarization switching? Or, is there any possibility to have a gradual rotation of the spin rotation plane or ferroelectric polarization?

Related to this issue, authors should provide a more discussion by comparing the RMCD and Pr obtained as a function of the magnetic field. In the RMCD loop, they observed a signature corresponding to spin flop process. According to the explanation given in the manuscript, I would expect a more dramatic variation of Pr at the corresponding magnetic field.

It will be useful to compare the polarization controllability authors observed for NiI2 with those of other multiferroics including CoI2. Also, there should be references cited for the statement "...~7%, ... which is remarkable feature of multiferroic." Actually, Ju et al. reported the change in the ferroelectric polarization for the 6L NiI2. (Nano Lett. 21, 5126 (2021)).

In the conclusion section, authors emphasized that NiI2 "will introduce a paradigm shift for engineering new ultra-compact magnetoelectric devices." I have to say this is over-stated. NiI2 may show the ME coupling in tri-layer, but this phenomenon can be observed only at the very low temperature. I think this statement diminish many other efforts to realize the ME coupling at room temperature.

There are several parts where English should be improved. Figures can be better described in the main text instead of repeating the text given in the figure caption. (Figure 4e, for example)

Overall, authors have discussed important issues about the magnetoelectricity in a tri-layer multiferroic NiI2. I would be willing to reconsider this work if authors can improve it reflecting my comments.

Reviewer #3

(Remarks to the Author)

Multiferroic materials displaying a strong magnetoelectric coupling are a promising platform for developing technological applications that require the control of magnetic orders using electric fields. Recently, evidence of multiferroic order has been reported in NiI₂ down to the few-layer [Hwiin Ju, et al Nano Letters 21, 5126–5132 (2021)] and monolayer limits [Qian Song, et al Nature 602, 601–605 (2022), Mohammad Amini, et al Advanced Materials, 2311342 (2024)]. These experiments have established NiI₂ as a new building block with huge potential for engineering van der Waals heterostructures with novel functionalities. However, proving multiferroicity in the monolayer limit by optical techniques is challenging [Jiang, Y. et al. Nature 619, E40-E43, (2023)] and new strategies need to be developed to demonstrate and characterize the multiferroic order in the low dimensional limit [Mohammad Amini, et al Advanced Materials, 2311342 (2024)]. Moreover, the evolution of the multiferroic order from the bulk to the single-layer limit remains unclear, and further studies need to be performed.

The authors of this work present an experimental analysis of a 3-layer NiI₂ device. Using scanning reflective magnetic circular dichroism microscopy and ferroelectric hysteresis loops, the authors show evidence of a non-collinear magnetic order and measure the out-of-plane component of the electric polarization in 3-layer NiI₂. Moreover, the authors show control of the ferroelectric polarization using an external magnetic field. From the experimental point of view, this study is timely and represents an important advance in the field of multiferroics and van der Waals materials. However, the theoretical interpretation of the experimental results shows some major inconsistencies that need to be amended before further consideration. If the authors address the following comments in the revised version of the manuscript, I will support the publication of this work in Nature Communications.

1) The authors identify the number of NiI₂ layers in their system using the Raman feature associated with the interlayer shear mode as explained in reference 20 of the manuscript. Could the authors also provide the optical contrast for the identification of the number of layers as also detailed in reference 20?

2) In caption (a) of Fig 2, the authors state that the RMCD maps are taken at room temperature, in the plot Fig 2a they show $T=10\text{K}$. I suppose that the latter is the right temperature of the maps, the authors should correct the caption.

3) Regarding the RMCD maps, Could the authors present in the SI more maps at different magnetic field values to show that the evolution with the magnetic field that they detail is robust?

4) The RMCD as a function of the magnetic field is shown in Fig 2d allowing us to identify a non-collinear magnetic order. It would be highly interesting if the authors could present results from 7T to 7T as they do for the electric polarization in Fig 4. This would help to clarify their interpretation of the decrease in polarization when applying an external magnetic field. One would think that if the spin spiral is aligned in the xy plane when an external magnetic field is applied parallel to the z direction, the RMCD signal should decrease since the RMCD intensity is proportional to the out-of-plane magnetization, which will go to zero if the spiral is aligned on the xy plane. However, if a transverse conical spin spiral is also formed when applying an external magnetic field the RMCD signal would increase while decreasing the out-of-plane electric polarization (see comments 7 and 8 for more details about the transverse conical spin spiral). The author should review this point and if possible show measurements of the RMCD signal from -7T to 7T as they do for the electric polarization.

5) In reference 42 the in-plane projection of the bulk q-vector along the $[210]$ (ie along the second-neighbor Ni atom direction) is theoretically considered to analyze the origin of the multiferroic order in NiI₂. However, it has been theoretically predicted (Sodequist et al 2D Materials 10, 035016 (2023)) and experimentally measured (Mohammad Amini, et al Advanced Materials, 2311342 (2024)) that the q in the monolayer goes along a different direction (third-neighbor Ni atom direction). For the 3-layer NiI₂ case that the authors present here, there is no evidence for the spin spiral q vector. It could be the q-vector shown in ref 42 if the bulk limit is already achieved in the 3-layer system (as suggested in ref 20), or a q-vector similar to the one found in the monolayer could arise. Since this limit is not explored yet, I would suggest the authors warn the reader about this point.

The theoretical interpretation that the authors provide for the decrease of electric polarization as a function of the external magnetic field presents inconsistencies.

6) First, the picture that they sketch from their results states that in the absence of a magnetic field, the spin spiral develops in the xz rotation plane with an e vector along the y direction producing an electric polarization in the z-direction. It has been theoretically shown that Kitaev interaction promotes a diagonal plane around 40 degrees for the spin spiral (Xuanyi Li, et al Phys. Rev. Lett. 131, 036701 (2023)). It would be more consistent with previous analyses and experimental observations to describe the magnetic state in the absence of an external magnetic field as a spin spiral in an oblique plane, rather than in the limiting xz plane.

7) Second, the authors propose that the external magnetic field modifies the plane of the spin spiral from xz to xy, which causes a reduction of the polarization along the z direction as shown by their results. However, this scenario should also cause a reduction of the RMCD signal as a function of the magnetic field, something at odds with the results presented in Fig 2d. The authors need to reconcile these two observations. The current theoretical interpretation is not consistent.

8) An alternative interpretation that I suggest the authors explore is the formation of a transverse conical spin spiral besides the reorientation of the spin spiral plane in the xy direction when applying the external magnetic field along the z-direction. This would decrease of the electric polarization in the z-direction while simultaneously increasing the magnetization along the z-direction as a function of the magnetic field in the z-direction. In this regard, I suggest the authors review (also improve the quality of the sketches) of Fig 7 in the SI as well as Fig 3 d,e,f to make the evolution of the magnetic order with the

magnetic field much clearer and consistent with all the experimental observations.

9) The references need to be highly improved. Relevant works in the field are missing: Hwiin Ju, et al Nano Letters²¹, 5126–5132 (2021) shows evidence of non-collinear magnetic order down to the bilayer limit using Second Harmonic Generation, Mohammad Amini, et al Advanced Materials, 2311342 (2024) demonstrates and characterizes the multiferroic order of monolayer NiI₂ using Scanning Tunneling Microscopy, Xuanyi Li, et al Phys. Rev. Lett. 131, 036701 (2023) introduces a realistic spin model to understand the magnetic properties of NiI₂, in particular the out-of-plane tilt of the spin spiral, Sodequist et al 2D Materials¹⁰, 035016 (2023) theoretically shows that in the monolayer the q vector of the spin spiral differs from the one in bulk. Some older, but also important references providing the first studies of the magnetic order in NiI₂ using neutron scattering: D. Billerey, et al Physics Letters A 61, 138–140 (1977) and S.R. Kuindersma, et al Physica B+C 111, 231–248 (1981).

10) There is a typo in the summary paragraph, “hysteresis loop” is repeated.

I hope that the authors can address all the comments described above to further support publication.

Version 1:

Reviewer comments:

Reviewer #1

(Remarks to the Author)

In the revised manuscript, the authors have done a lot of additional experimental works

(such as SHE, LD and temperature dependent Raman spectra) and performed the DFT

calculations as well as the atomistic spin model simulations. The responses have

satisfactorily addressed most of my previous comments. I just have few minor comments.

1. The author mentioned that the observed background current in Fig. 4e should be

identified as displacement current rather than conduction current. This is an important

information. I suggest the author to include related discussion in the main text or

Supplementary Materials. Also, maybe the authors can give the DC measurement of the

resistance of the devices (including Gr/NiI₂/Gr, Gr/hBN/Gr and Gr/NiI₂/hBN/Gr,) to

prove this conclusion.

2. The P-V and I-V loops from the Gr/NiI₂/Gr device give solid evidence that the

introduction of the insulating hBN layer can effectively reduce the leakage current. I

suggest the authors to include these figures in the Supplementary Materials.

3. I think the discussion on the Raman scattering intensity based on the Raman selection rule will be helpful to the readers who are not very familiar with Raman measurements. I

suggest the authors to include the discussion in Supplementary Fig. 2.

Reviewer #2

(Remarks to the Author)

Authors replied comments and revised the manuscript satisfactorily. I have one thing to ask the authors to implement in the manuscript. Although the DFT calculations could give information about the q-vector, it was not experimentally verified.

Therefore, authors should be more careful when they discuss the magnetoelectricity based on the given q-vector. Authors may explicitly state that the q-vector is not experimentally verified.

Overall, this work clearly demonstrates the magnetoelectricity of the tri-layer NiI₂ which is worthy to be published in Nature Communications. I strongly recommend its publication.

Reviewer #3

(Remarks to the Author)

The authors have addressed all the comments raised in the previous report and modified the article accordingly. The inconsistencies observed in the previous version have been successfully resolved and the new version of the manuscript has been substantially improved. The work is timely and represents an important advancement in the fields of van der Waals

materials and multiferroics. Therefore, I support the publication of the current version of this article in Nature Communications.

Version 2:

Reviewer comments:

Reviewer #1

(Remarks to the Author)

The authors have modified the manuscript accordingly. I now believe publication in Nature Communications is warranted.

RESPONSE TO REVIEWERS' COMMENTS

Reviewer #1 (Remarks to the Author):

In the manuscript entitled “Coexistence of ferroelectricity and antiferroelectricity in 2D van der Waals multiferroic” by Y. L. Wu et al., the authors reported the direct electrical measurements of spontaneous polarization in 2D vdW NiI₂ trilayers. By analysing the P-E and I-E hysteresis loops, the author concluded that both ferroelectric and antiferroelectric phases coexisted in the NiI₂ trilayers. And the out-of-plane ferroelectric polarization can be tuned by an out-of-plane magnetic field. The result and the interpretation look interesting to me. However, I notice that the multiferroicity order in NiI₂ monolayer or few layers has been extensively studied in recent years (10.1038/s41586-021-04337-x, 10.1088/2053-1583/ac4e9d, 10.1002/adma.202311342, 10.1002/adma.202109144, 10.1021/acs.nanolett.1c01095, 10.1021/acs.nanolett.1c01095, 10.1103/PhysRevLett.131.036701). Though the statement of antiferroelectric polarization would be a new finding, it lacks sufficient experimental or theoretical support in the present manuscript. Therefore, the novelty of this manuscript should be further clarified. And there are also some questions that the authors need to address:

Reply: We sincerely thank you for your interest in our research and the considerable time and effort you dedicated to evaluating our work. In response, we have emphasized the uniqueness and novelty of our manuscript by thoroughly comparing it with the recommended literature. Furthermore, we carried out additional DFT simulations, atomistic spin model simulations, and modeling calculations to validate the magnetic structures of trilayer NiI₂ and conducted supplementary experiments to address the primary concerns. Your valuable suggestions have significantly improved our manuscript.

Recent literature has assumed that type-II multiferroicity probably persists even in the few-layer and monolayer limit (*Nano letters* **21**, 5126-5132, 2021; *Nature* **602**, 601-605, 2022). To validate 2D ferroelectricity, linear dichroism (LD) and second-harmonic generation (SHG) measurements were performed on both monolayer and multilayer NiI₂.

However, it has been noted in a recent Nature paper that LD and SHG are inadequate to confirm the ferroelectric nature of 2D NiI₂ (*Nature* 619, E40-E43, 2023). SHG can be evidently observed in any system that lacks ferroelectricity but exhibits inversion symmetry breaking (*Nature* 572, 497-501, 2019; *Nat. Phys.* 14, 229-241, 2018; *J. Opt. Soc. Am. B* 22, 148-167, 2005). Importantly, SHG can manifest in a variety of magnetic materials due to inversion symmetry breaking induced by magnetic ordering, without necessitating ferroelectric properties (*npj Quantum Mater.* 24, 62, 2023). In magnetic systems, the use of LD as evidence for ferroelectricity is particularly challenging and often unreliable, as strong LD signals can emerge in purely magnetic materials, independent of any ferroelectric contribution (*Nat. Rev. Phys.* 1, 646-661, 2019; *Phys. Rev. Lett.* 83, 1862, 1999; *Phys. Rev. Lett.* 94, 227203, 2005; *Commun. Mater.* 1, 39, 2020).

Alternatively, scanning tunneling microscopy (STM) has been applied to detect spatial fluctuations of the tunneling current in monolayer NiI₂, and periodic stripes of 1.78 nm have been observed, which were speculated to result from ferroelectric polarization (*Adv. Mater.* 36, 2311342, 2024). **However, in stark contrast, the spin-polarized STM (SPSTM) results have demonstrated that the observed periodic tunneling current stripes in monolayer NiI₂ stem from helical magnetic orders rather than ferroelectric orders. Moreover, the period is 1.76 nm, very closely matching the theoretically calculated 1.77 nm period of the magnetization density wave induced by helical magnetic orders ([arXiv:2309.16526](https://arxiv.org/abs/2309.16526)).** Therefore, convincing experimental evidence for the multiferroic properties of few-layer NiI₂ remains challenging and elusive.

As shared by Reviewers 2 and 3, this work, by employing appropriate and reliable methods, provides 'smoking-gun' evidence to verify the multiferroicity of trilayer NiI₂. It also demonstrates significant magnetoelectric coupling at the limit of few atomic layers. Moreover, the observed unusual out-of-plane ferroelectricity is newly discovered and breaks through the previously common sense that NiI₂ is an in-plane ferroelectric material. Compared to in-plane ferroelectricity, out-of-plane ferroelectricity offers greater potential advantages in information storage. Our theoretical calculations reveal that this out-of-plane ferroelectricity originates

from the unique tilted spin-rotation plane in trilayer NiI₂, significantly enhancing the understanding of physical properties in the 2D type II multiferroics field. In addition to the ferroelectric phase, an intriguing antiferroelectric phase has been observed, with atomistic spin model simulations revealing its origin in the chiral-degenerate spiral magnetic order.

Our work evidently differs from and goes beyond the previous works mentioned by the respected reviewer, as explained in detail by the following points.

1. Ju, H. et al. *Possible persistence of multiferroic order down to bilayer limit of van der Waals material NiI₂*. *Nano letters* **21**, 5126-5132 (2021).

Remark: This earliest work suggests that SHG indicates the **possible** persistence of the multiferroic properties in a few layers of NiI₂. We appreciate the authors' rigorous scholarship, as they noted that the SHG is not convincing and ferroelectricity may not be present. **In our work, we demonstrate convincing evidence of multiferroicity, as revealed by scanning reflective magnetic circular dichroism microscopy and the ferroelectric hysteresis loop.**

2. Song, Q. et al. *Evidence for a single-layer van der Waals multiferroic*. *Nature* **602**, 601-605, (2022).

Remark: In fact, Ref. 2 is similar to Ref. 1, but the authors of Ref. 2 have introduced significant pitfalls for general readers, leading to confusion in the field of 2D multiferroic materials. In Ref. 2, SHG and LD were employed to characterize the multiferroic properties of few-layer NiI₂. However, SHG and LD cannot be considered sufficient evidence for judging multiferroicity, as indicated by the rigorous scientific wording '**possible evidence**' in the title of Ref. 1. **The LD and SHG are inadequate to identify ferroelectricity in 2D NiI₂** (*Nature* **619**, E40-E43, 2023; *npj Quantum Mater.* **24**, 62, 2023; *Nat. Rev. Phys.* **1**, 646-661, 2019; *Nat. Photon.* **16**, 311-317, 2022; *Phys. Rev. Lett.* **83**, 1862, 1999; *Phys. Rev. Lett.* **94**, 227203, 2005). **Remarkably, our work, by employing appropriate and reliable magneto-optical-electric joint-measurement methods, provides 'smoking-gun' evidence to verify the multiferroicity of trilayer NiI₂ and also demonstrates**

significant magnetoelectric coupling. Moreover, our theoretical calculations reveal that this out-of-plane ferroelectricity originates from the unique tilted spin-rotation plane in trilayer NiI₂, significantly enhancing the understanding of physical properties in the 2D type II multiferroics field.

3. Amini, M. et al. Atomic-scale visualization of multiferroicity in monolayer NiI₂. Adv. Mater. 36, 2311342, (2024).

Remark: Amini et al. reported periodic spatial fluctuations in the tunneling current of monolayer NiI₂ using scanning tunneling microscopy (STM), with a period of 1.78 nm, and considered them indicative of ferroelectric polarization. **However, in stark contrast**, Miao et al. observed similar periodic tunneling current patterns in monolayer NiI₂ using **spin-polarized STM (SPSTM)**, with a period of 1.76 nm, which is well consistent with the theoretical calculations of the magnetization density wave associated with helical magnetic order. Crucially, SPSTM results suggest that the periodic tunneling current patterns originate from helical magnetic order rather than ferroelectric effects (*arXiv:2309.16526*). Therefore, the current experimental data are in debate for providing compelling support for the multiferroic properties of few-layer or monolayer NiI₂. **Particularly, the nature of magnetoelectric coupling in the few-layer limit remains ambiguous.**

4. Son, S. et al. Multiferroic-enabled magnetic-excitons in 2D quantum-entangled Van der Waals antiferromagnet NiI₂. Adv. Mater. 34, 2109144, (2022).

Remark: The research focuses on the magnetic exciton in bulk NiI₂ and does not explore the multiferroic properties in the few-layer or monolayer limits.

5. Fumega, A. O. et al. Microscopic origin of multiferroic order in monolayer NiI₂. 2D Mater. 9, 025010, (2022).

Remark: This literature provides only a theoretical explanation for the multiferroic properties in monolayer NiI₂. However, the multiferroic properties of monolayer NiI₂ at low temperatures below 20 K remain challenging and represent an unresolved issue

due to the absence of successful direct experimental measurements. Notably, **our work demonstrates convincing experimental evidence of multiferroicity, as revealed by scanning reflective magnetic circular dichroism microscopy and the ferroelectric hysteresis loop.**

6. Li X. Y. et al. Realistic spin model for multiferroic NiI₂. Phys. Rev. Lett. 131, 036701 (2023).

Remark: This work theoretically predicted the presence of helical magnetism in NiI₂ and reported that the rotation plane and propagation vector of the helical magnetism are influenced by the Kitaev interaction. **However, the conclusive experimental evidence of the multiferroic properties of few-layer NiI₂, particularly the magnetoelectric coupling effect, have not yet been revealed. Importantly, our work provides convincing experimental evidences of multiferroicity in trilayer NiI₂.**

1) The Raman spectra of the NiI₂ device only show two peaks at 124.7 cm⁻¹ and 20 cm⁻¹. At what temperature does the author perform the Raman spectra? Why the Raman peak of E_g symmetry around 75~80 cm⁻¹ (as reported in previous works 10.1038/s41586-021-04337-x, 10.1021/acsnano.0c04499) can not be identified in the Raman spectra? I also suggest the author to measure the temperature-dependent Raman spectra for their NiI₂ trilayers. Whether the Raman response in the magnetically ordered phase can be seen at low temperatures. This is important for confirming the AFM order in their NiI₂ devices.

Reply: We thank you for the helpful suggestion. Circular polarization-resolved Raman spectra of a trilayer NiI₂ device were obtained at room temperature (Fig. 1d). The detection of Raman modes is closely related to the Raman tensor and the polarization configurations of the incident and scattered light. According to the Raman selection rule, the Raman scattering intensity is proportional to $|\sigma_s^\dagger \cdot \tilde{R} \cdot \sigma_i|^2$. For rhombohedral ($R\bar{3}m$) NiI₂, the A_{1g} and E_g modes are active. The Raman intensities of A_{1g} (124.7 cm⁻¹) and E_g (76 cm⁻¹) in circularly polarized configurations of $\sigma^+\sigma^+$ and $\sigma^-\sigma^-$ channels

are

$$I_{A_{1g}}^{\sigma-\sigma^-} = I_{A_{1g}}^{\sigma+\sigma^+} \propto a^2$$

$$I_{E_g}^{\sigma-\sigma^-} = I_{E_g}^{\sigma+\sigma^+} = 0$$

Hence, in the $\sigma+\sigma^+$ and $\sigma-\sigma^-$ channels, only A_g is detectable, whereas E_g is not detectable, aligning well with the Raman spectra (Fig. 1d).

Alternatively, in the linearly polarized configurations of XX and XY channels, the Raman intensities of the A_{1g} and E_g modes are

$$I_{A_{1g}}^{XX} \propto a^2 \quad I_{A_{1g}}^{XY} = 0$$

$$I_{E_g}^{XX} \propto c^2 \quad I_{E_g}^{XY} \propto c^2$$

Therefore, the E_g mode is observed in both XX and XY channels, while the A_{1g} mode is only identifiable in the XX channel. This observation is consistent with the results obtained in our Raman experiments (see Fig. R1), which are also consistent with previous researches. (*ACS Nano* **14**, 10544-10551, 2020; *Nature* **602**, 601-605, 2022).

Fig. R1. Raman spectra of trilayer NiI₂ were obtained in channels XX and XY at room temperature.

To investigate the relationship between magnetic order and phonon structure, we have conducted variable-temperature micro-Raman spectroscopy using co-polarized channels ($\sigma+\sigma^+$ channel) (Fig. R2a). At 10 K, a peak at approximately 76 cm⁻¹ was detected. Additionally, a new peak at around 30 cm⁻¹, attributed to the magnon mode

and labeled as M, was also observed (*ACS Nano* **14**, 10544-10551, 2020). As the temperature increases, there is a gradual decrease in the intensity of the peaks observed at 78 cm^{-1} and 30 cm^{-1} , suggesting a transition temperature of approximately 35 K (Fig. R2b). This observation is consistent with the established magnetic phase transition temperature reported for trilayer NiI_2 in previous studies (*arXiv:2307.10686*, 2023; *Nano Lett.* **21**, 5126-5132, 2021; *ACS Nano* **14**, 10544-10551, 2020).

Fig. R2. **a**, Circular polarization-resolved Raman spectra of trilayer NiI_2 at temperatures ranging from 10 K to 45 K. **b**, Temperature-dependent Raman peak intensities of the E_g and M modes, indicating a transition temperature of magnetic orders.

2) The direction of applied magnetic field for the RMCD imaging should be clearly pointed out in the figures. The RMCD sweeping loops in Fig. 2d and Fig. S2b are obtained from the average value of whole mapping area or just from a local area? If the latter, the special area should be marked in the mapping image.

Reply: We extend our gratitude to you for your valuable comments. The orientation of the applied magnetic field is depicted by arrows in the RMCD imaging (Fig. 2a), aligned parallel to the c -axis. The RMCD loops presented in Fig. 2d and Supplementary Fig. 3b were collected from specific local areas, which have been identified by black circles in the RMCD imaging (middle panel in Fig. 2a and Supplementary Fig. 3j).

Additionally, more RMCD maps have been added in the revised manuscript (Supplementary Fig. 3).

3) The author said that "The RMCD intensity near 0 T is suppressed and approaches zero, suggesting the vanishing remnant magnetization, which indicates a compensation of the out-of-plane magnetization and non-collinear AFM orders in the trilayer NiI₂" I think this can not be sufficient evidence for the AFM order in the NiI₂ trilayers. It could be also due to the strong in-plane anisotropy, which becomes dominant at low magnetic field. So more evidence for the AFM order is needed.

Reply: We express our gratitude to you for the valuable suggestions regarding the magnetic ground state of trilayer NiI₂. We acknowledge that determining the antiferromagnetic (AFM) order of trilayer NiI₂ solely from the RMCD loops remains challenging. The AFM orders are synthetically deduced based on the theoretical and experimental results:

(1) To confirm the magnetic ground state of trilayer NiI₂, density functional theory (DFT) calculations have been performed, revealing a spiral spin configuration with an in-plane projection of the \mathbf{q} -vector along the [210] direction (indicated by the pink arrow in Fig. R3d). The magnetic order is illustrated only for the top Ni layer for clarity, while the magnetic moments of the 3L Ni are presented in Fig. R4. Trilayer NiI₂ exhibits interlayer AFM coupling due to the larger second interlayer nearest-neighbor exchange parameter (as shown in Fig. R3e). The canting of the rotation plane is predominantly determined by the Kitaev interaction (*Phys. Rev. Lett.* **131**, 036701, 2023; *Phys. Rev. B.* **109**, 195422, 2024), with the in-plane projection of the \mathbf{e} -vector perpendicular to the direction of \mathbf{a} (indicated by the green arrow in Fig. R3d).

(2) Atomistic spin model simulations (see the supplementary materials) have been conducted and demonstrate that spiral spin states give rise to stripy domains (Fig. R3f), suggesting that the spiral spin state is the ground state of trilayer NiI₂, which is consistent with the Monte Carlo simulations (*Phys. Rev. Lett.* **131**, 036701, 2023). To clearly show the magnetic structure, Fig. R3f displays only the spins of the initial Ni layer, while the spin textures of the other two layers are presented in Fig. R5a and c. The

interlayer AFM coupling induces opposite magnetic moment components in adjacent layers.

(3) Moreover, variable-temperature Raman spectroscopy has been performed in co-polarized channels ($\sigma+\sigma+$ channel) (Fig. R2a). At 10 K, a peak around 76 cm^{-1} was observed, along with a new peak at approximately 30 cm^{-1} , attributed to the magnon mode, denoted as M. As the temperature rises, the peaks at 76 cm^{-1} and 30 cm^{-1} gradually diminish, with a transition temperature of around 35 K (Fig. R6a), consistent with the reported transition temperature of AFM orders in trilayer NiI_2 (*ACS Nano* **14**, 10544-10551, 2020; *Nano Lett.* **21**, 5126-5132, 2021). The close link between the transition temperatures observed in the Raman and SHG results originates from the phase change of non-collinear antiferromagnetic orders (Fig. R6b), supporting the presence of non-collinear antiferromagnetism in trilayer NiI_2 .

In summary, the comprehensive experimental and theoretical evidence effectively supports the conclusion that trilayer NiI_2 exhibits non-collinear antiferromagnetic orders.

Fig. R3. **a, b**, P - E and I - E loops at various frequencies from device 1 (D1). **c**, Corresponding I - E loops from Fig. R3b subtracted the current background. Two pairs of current peaks (FE-AFE and AFE-FE switching peaks) were obtained by Lorentz fitting. An evolution from FE to AFE was observed. **d**, Schematics of spiral configuration in top views. **e**, Schematics of spiral configuration in side views, showing the interlayer AFM coupling and out-of-plane ferroelectricity. **f**, Spin texture of NiI₂ from atomistic spin model simulations. Spins are represented by arrows, with the red and blue colors showing positive and negative values of the S_z component. **g**, The corresponding dipole texture obtained through calculations using the KNB model. Dipoles are represented by arrows, with the red and blue colors showing positive and negative values of the P_z component. **h**, An enlarged view of the out-of-plane AFE domain occurring in panel **g** (light blue solid line box).

Fig. R4. Top views of the magnetic orders of trilayer NiI_2 . The blue, green, and gray balls represent Ni in the first, second, and third layers, respectively, while the arrows indicate the spins of Ni.

Fig. R5. **a, c**, Spin textures and corresponding dipole textures (**b, d**) from numerical simulations. Spins and dipoles are represented by arrows, with red and blue colors indicating positive and negative values of the out-of-plane component. The scale bars are 2 nm.

[Figure redacted]

Fig. R6. a, Temperature-dependent Raman peak intensities of the E_g and M modes (magnon scattering), respectively. **b**, Temperature-dependent second-harmonic generation (SHG) intensity (*Nano Lett.* **21**, 5126-5132, 2021).

4) In Fig. 3, the author claimed only a single peak were observed when sweeping voltage at 6.7 Hz. This is not true. The broad shoulder peak around $+0.1 \text{ MVcm}^{-1}$ for the I-E curve at 6.7 Hz is very similar to the I-E curves at 3.3 Hz. Moreover, why the FE-AFE transition is more significant at low frequency needs further explanation.

Reply: We appreciate you for scrutinizing our manuscript. **The vital point is that the evolution of current features with increasing frequency suggests a transition from antiferroelectricity (double peak) to ferroelectricity (single peak), and the E_{AFE-FE} current features gradually decrease and disappear as the frequency increases from 1.3 to 6.7 Hz.** Our experimental results have evidently shown the evolution, which strongly validates the coexistence of antiferroelectricity and ferroelectricity. The E_{AFE-FE} current peak at 6.7 Hz nearly disappears and is very hard to observe. As depicted in Fig. R7a, only the FE current peak is detected, and it is consistent with a single-peak fitting. However, when reducing the frequency to 3.3 Hz, a single-peak model **cannot** fit the feature well, and a distinct discrepancy appears in the region marked by the black dashed circle (Fig. R7b). In stark contrast, a double-peak fitting provides an accurate representation of the experimental data (Fig. R7c). At 1.3 Hz, the presence of a double current peak becomes more prominent (Fig. R3c). These observations suggest that a

gradual transition from antiferroelectricity (double peak) to ferroelectricity (single peak) as the frequency increases. In accordance with the KNB or inverse D-M mechanism (*Phys. Rev. Lett.* **29**, 057205, 2005; *Phys. Rev. B* **73**, 094434, 2006), the dipole ordering configuration can be deduced from the spiral spin texture (Fig. R3g and Figs. R5b, d). Computational simulations have revealed the coexistence of ferroelectric domains with opposite P_z components, attributed to the distinct chirality of the magnetic domains (highlighted by the areas enclosed by the red and black dashed lines in Fig. R3g), because the spiral magnetic states with opposite chirality are degenerated but with opposite ferroelectric polarization (*2D Mater.* **9**, 025010, 2022; *Phys. Rev. B* **109**, 195422, 2024). Remarkably, in certain transitional regions, local dipoles with opposing P_z components are interwoven, which induces antiferroelectricity in the out-of-plane direction (Fig. R3h). The calculated intricate intermingling of ferroelectric and antiferroelectric domains strongly supports the experimentally observed coexistence of ferroelectricity and antiferroelectricity. The evolution of the ferroelectric hysteresis loop with frequency is attributed to the distinct reversal dynamics between ferroelectric and antiferroelectric domains. Notably, similar frequency-dependent behaviors have been observed in complex multi-domain systems (*Ceram. Int.* **45**, 20276-20281, 2019).

Fig. R7. **a**, single-peak-fitting at 6.7 Hz. **b**, single-peak-fitting at 3.3 Hz. **c**, double-peak-fitting at 3.3 Hz.

5) Though the author concluded the electric measurement was more reliable than the general SHG and LD measurement, I think they should also measure the SHG and LD signals for their NiI₂ devices. This comparison between SHE, LD and to P-E/I-E loops may give us some hints on the advantages of direct electric measurement.

Reply: We thank you for the suggestion. To compare magnetic domains with LD optical domains, we have performed RMCD and LD measurements on few-layer NiI₂, as depicted in Figs. R8a-e. RMCD mapping is a well-established technique for detecting magnetic domains (*Science* **374**, 1140-1144, 2021; *Nat. Mater.* **18**, 1303-1308, 2019). Our experimental results reveal a strong correlation between the RMCD and LD maps, suggesting that the observed LD signals stem from the magnetic orders. Notably, the LD optical domains remain unchanged under an external electric field (Figs. R8b-e), ruling out the possibility of the LD signal originating from ferroelectricity. This further validates that LD signals arise from magnetic order rather than ferroelectric polarization. Moreover, the SHG signals have been demonstrated to originate from the magnetic orders rather than ferroelectric polarization (*arXiv:2307.10686*, 2023). Alternatively, the non-ferroelectric antiferromagnetic (AFM) CrI₃ exhibits strong SHG signals. These SHG signals are exclusively associated with the AFM order and disappear in ferromagnetic (FM) states (Figs. R8f and g). This is because AFM magnetic orders lead to the breaking of inversion symmetry. Therefore, there is no direct relationship in ferroelectric polarization. The relations between SHG, LD and P-E loops are extraordinarily complex. And it is very challenging to clearly clarify and distinguish them on theoretical and experimental studies. Thus, at present, P-E loops as a reliable method are independently used to identify the ferroelectric polarization of multiferroic NiI₂.

[Figure redacted]

Fig. R8. Experimental results of RMCD (**a**) and LD (**b-e**) intensity images of a few-layer NiI_2 obtained at the same spot under different electric fields at 10 K for comparison. Scale bars are 1 μm . **f**, SHG intensity of CrI_3 bilayer as a function of magnetic field. The magnetic field was swept downwards (red) and upwards (blue). **g**, Corresponding RMCD hysteresis loop (*Nature* **572**, 497-501, 2019).

6) The author claimed that the easy axis would tilt from the x-y plane of the NiI_2 monolayer to the x-z plane of the NiI_2 trilayer, and an out-of-plane magnetic field can rotate the spin back to x-y plane. However, these are all hypothesis, lacking of either experimental evidence of theoretical calculations.

Reply: We thank you for this suggestion. This question is closely related to Question 3 above. In the original manuscript, it is proposed that the out-of-plane ferroelectric polarization emerges because the spin rotation plane is tilted outward in trilayer NiI_2 , rather than lying entirely in the xy plane. This is consistent with recent theoretical reports on helical magnetic orders with a canted spin-rotation plane in NiI_2 (*Phys. Rev. Lett.* **131**, 036701, 2023; *Phys. Rev. B* **109**, 195422, 2024).

To confirm the magnetic ground state of trilayer NiI_2 , DFT calculations have been performed, demonstrating that the magnetic ground state exhibits a spiral spin

configuration with an in-plane projection of the \mathbf{q} -vector along the [210] direction (as indicated by the pink arrow in Fig. R3d). For clarity, the magnetic order is illustrated for only the top Ni layer, while the magnetic moments of the 3L Ni are presented in Fig. R4. Trilayer NiI₂ exhibits interlayer antiferromagnetic (AFM) coupling due to the larger second interlayer nearest neighbor exchange parameter (Fig. R3e). The canting of the rotation plane is predominantly determined by the Kitaev interaction, and the in-plane projection of the \mathbf{e} -vector is perpendicular to the direction of \mathbf{a} (as indicated by the green arrow in Fig. R3d). Therefore, a ferroelectric polarization emerges from the spiral magnetic order (*Phys. Rev. Lett.* **29**, 057205, 2005; *Phys. Rev. B* **73**, 094434, 2006; *Nat. Mater.* **6**, 13-20, 2007; *Phys. Rev. Lett.* **96**, 067601, 2006), and the geometrical arrangement of \mathbf{q} and \mathbf{e} results in the out-of-plane component (c-axis direction) of ferroelectric polarization (Fig. R3e), which is consistent with the experimentally observed out-of-plane ferroelectricity (Fig. R3a).

Furthermore, the in-plane and out-of-plane magnetic field control of ferroelectric polarization experimentally validates the DFT results, which are demonstrated in detail in the next reply.

7) If the magnetic field is applied in-plane, what will happen to the ferroelectric properties of NiI₂ device. According to the hypothesis of spin easy plane, the in-plane field would not change the P?

Reply: The in-plane magnetic field can also modulate ferroelectric polarization.

We are very sorry for that our writing and explanations are not friendly and hinder readers' understanding. In our revised manuscript, the models and explanations have been revised. As shown in Fig. R9 (Supplementary Fig. 11), the ferroelectric behavior in spiral magnets is elucidated by the inverse D-M interaction, expressed as $\mathbf{P} \propto \mathbf{e} \times \mathbf{q}$, where \mathbf{q} represents the spiral's propagation vector and \mathbf{e} denotes the rotational axis of the spin-rotation plane. In the top panel of Fig. R9, the spins within spiral magnets rotate in the ac -plane ($\mathbf{e} \parallel \mathbf{b}$), forming a transverse spiral with $\mathbf{q} \parallel \mathbf{a}$, resulting in $\mathbf{P} \parallel \mathbf{c}$. The middle panel illustrates that an applied magnetic field $\mathbf{H} \parallel \mathbf{c}$ induces magnetization

$\mathbf{M} // \mathbf{H} // \mathbf{c}$ through additional Zeeman energy, leading to a transverse conical spin configuration with an effective ab rotation plane where $\mathbf{e} // \mathbf{H} // \mathbf{c}$ (*Phys. Rev. Lett.* **105**, 187601, 2010; *Phys. Rev. Lett.* **96**, 067601, 2006). Consequently, a reorientation of \mathbf{P} from $\mathbf{P} // \mathbf{c}$ to $\mathbf{P} // \mathbf{b}$ takes place. This scenario is consistent with experimental results showing that an out-of-plane magnetic field induces out-of-plane magnetization (Fig. 2d) and decreases out-of-plane ferroelectric polarization (Fig. 4a-c). When an in-plane magnetic field is applied along a -axis ($\mathbf{H} // \mathbf{q} // \mathbf{a}$), a longitudinal conical spin configuration with an effective bc rotation plane is established, as depicted in the bottom panel of Fig. R9. In this scenario ($\mathbf{e} // \mathbf{q}$), the ferroelectric polarization is expected to be zero according to the inverse D-M model ($\mathbf{P} \propto \mathbf{e} \times \mathbf{q}$). Overall, the magnetic field aligns the \mathbf{e} -vector to the direction of applied magnetic field, leading to the reorganization of ferroelectric polarization.

Importantly, the in-plane magnetic field has been applied to validate the magnetic control behaviors of ferroelectric polarization. The in-plane magnetic fields oriented parallel and perpendicular to the \mathbf{S} -vector (sample orientation), as illustrated in Figs. R10a and c. Strikingly, for $\mathbf{H} // \mathbf{S}$, the out-of-plane ferroelectric polarization decreases with the magnetic field increase (Fig. R10b). Conversely, for $\mathbf{H} \perp \mathbf{S}$, the ferroelectric polarization increases with increasing magnetic field h (Fig. R10d). In the trilayer NiI_2 , the magnetic ground state adopts a spiral configuration with a propagation vector \mathbf{q} along the $[210]$ direction (indicated by the pink arrow in Fig. R10e, f), while the \mathbf{e} -vector is perpendicular to the a -axis, denoted by the green arrow. When the in-plane magnetic field is oriented between the \mathbf{e} -vector and the \mathbf{q} -vector, it harnesses the \mathbf{e} -vector to align to the magnetic field direction, leading to a decreasing angle between the \mathbf{e} -vector and the \mathbf{q} -vector, consequently diminishing the ferroelectric polarization (Figs. R10a, b and g). Conversely, when the magnetic field is perpendicular to the \mathbf{S} -vector, it causes an increase in the angle between the \mathbf{e} -vector and the \mathbf{q} -vector, resulting in an enhancement of the ferroelectric polarization (Figs. R10c, d and h). The experimental results are well consistent with the magneto-electric coupling mechanism.

Fig. R9. Schematics of ac -plane transverse spin spiral with propagation vector $\mathbf{q} \parallel \mathbf{a}$ (top panel), which induces ferroelectric polarization $\mathbf{P} \parallel \mathbf{c}$. Application of magnetic field ($\mathbf{H} \parallel \mathbf{c}$, middle panel) and ($\mathbf{H} \parallel \mathbf{a}$, bottom panel) is expected to stabilize the transverse and longitudinal spin conical with magnetization $\mathbf{M} \parallel \mathbf{H}$, respectively, where $\mathbf{P} \parallel \mathbf{a}$ and $\mathbf{P} = 0$ is expected within the KNB model.

Fig. R10. **a, c**, Optical photographs of the sample at different geometric configurations where the magnetic field is parallel or perpendicular to \mathbf{S} -vector (sample orientation). The direction of the magnetic field is fixed, and the parallel ($\mathbf{H} \parallel \mathbf{S}$) and perpendicular configurations ($\mathbf{H} \perp \mathbf{S}$) are achieved by rotating the sample. **b, c**, P_r extracted from the P - E hysteresis loop as a function of in-plane magnetic field at $\mathbf{H} \parallel \mathbf{S}$ and $\mathbf{H} \perp \mathbf{S}$. **e**, Schematics of spiral configuration in top views. **f, g, h**, Magnetic-field control behavior of \mathbf{e} vs \mathbf{q} .

8) *The antiferroelectric phase is interesting finding. Can the author give more provement?*

Reply: We thank you for the positive evaluation of our finding and for the helpful suggestion. The evolution of the capacitance-voltage (C - V) characteristics with frequency at 10 K was meticulously analyzed. At 6.7 Hz, the C - V curve exhibited a distinct single-butterfly shape (top panel in Fig. R11a), indicating prominent ferroelectric characteristics (*Nat. Commun.* **3**, 1064, 2012). Strikingly, at 1.3 Hz, the C - V curve demonstrated characteristic antiferroelectric behavior with a double-butterfly shape, further confirming the antiferroelectric performance (bottom panel in Fig. R11a). The observation of pinched double P - E loops and double-butterfly C - V curves in few-layer NiI_2 closely resembles the antiferroelectric features seen in classical PbZrO_3 (Figs. R11b and c), providing strong evidence for the antiferroelectric properties of few-layer NiI_2 (*Nat. Commun.* **12**, 4215, 2021; *Nat. Commun.* **4**, 2229, 2013; *Nat. Commun.* **15**, 3438, 2024). Notably, DFT, atomistic spin model simulations, and numerical calculations have demonstrated that the local dipoles with opposing P_z components are interleaved, serving as the underlying mechanism for antiferroelectricity (Fig. R3h).

Fig. R11. **a**, Capacitance-voltage (C - V) characteristics measured at 6.7 Hz (top panel) and 1.3 Hz (bottom panel) for the trilayer NiI_2 . **b**, P - E loop of PbZrO_3 (PZO) film. **c**, Capacitance-voltage (C - V) characteristics measured at 100 kHz for the Pt/PZO/LSMO capacitor (*Nat. Commun.* **12**, 4215, 2021).

9) *I don't understand the conclusion "The shifts of current peaks induced by ferroelectric switching vary with the magnetic field, but the background current remains constant, excluding the magnetoresistance effects". In my opinion, the resistance of NiI_2*

layer should be different at zero field and at a very high field of 7 T. So the different background current is the normal case. Please give the magnetoresistance vs. H curves for the NiI₂ device at both 10 K and 300 K, to confirm the unchanged magnetoresistance below T_N.

Reply: We express our gratitude for the meticulous review of our paper and the valuable feedback provided. We respectfully present a different viewpoint on the background current. The observed background current (Fig. 4e) is identified as displacement current rather than conduction current, thus no phenomenon can be detected associated with conduction current, such as Joule heating and magnetoresistance effects. Displacement current is defined as the time rate of change of the electric displacement flux passing through a surface, a concept initially introduced by Maxwell. The displacement current density vector is symbolized as $\mathbf{j} = \partial \mathbf{D} / \partial t$. Considering the relationship $\mathbf{D} = \epsilon_0 \mathbf{E} + \mathbf{P}$, where \mathbf{E} represents the electric field vector and \mathbf{P} denotes the polarization vector in dielectric materials, the displacement current density \mathbf{j} is expressed as $\mathbf{j} = \epsilon_0 \partial \mathbf{E} / \partial t + \partial \mathbf{P} / \partial t$. Consequently, for dielectrics polarized linearly with the electric field, such as hBN, the background current remains constant and escalates with the frequency increase (i.e., the rate of change of the electric field over time) (Supplementary Fig. 5c in the revised manuscript). In stark contrast, in ferroelectric materials, the ferroelectric polarization undergoes a sudden reversal at the critical electric field (large $\partial \mathbf{P} / \partial t$), leading to a substantial current peak (Figs. R3b and c). In Fig. 4e, the magnetic field solely influences the current peak, suggesting that the magnetic field governs the ferroelectric polarization reversal with the electric field in NiI₂. However, the background current does not participate in the ferroelectric polarization reversal; hence, the magnetic field does not impact the background current. Moreover, the background current is non-conductive, there is no magnetoresistance effect. The experimental results derived from the graphene/NiI₂/hBN/graphene device demonstrate that the background current arises from the capacitor's charging and discharging process, rather than the conductive tunneling current through the NiI₂ layer. Consequently, the ferroelectric measurement does not involve magnetoresistance.

10) Fig. S2c-e show the RMCD maps for a few-layer NiI_2 . Except for the bimerons-like domains, the RMCD signal in other area is inconsistent with the RMCD sweeping in Fig. S2b. For example, positive signal is observed at 0 T in d, and the zero signal is observed at -0.75 T in e.

Reply: The RMCD maps are consistent with the RMCD loops. The precise numerical value in the RMCD image is challenging to distinguish visually. Figure R12a shows that the RMCD intensity at -0.75T is approximately -0.000602. The RMCD collection position is indicated by a dashed circle in Fig. R12c. And the corresponding RMCD intensity collected in RMCD maps is around -0.000623 (Fig. R12c), which is very close to RMCD loop intensity. Figure R12b illustrates the RMCD signal intensity obtained by scanning along the dashed line in the RMCD maps. Apart from the bimeron-like domains, the RMCD signal at other positions shows slight fluctuations around -0.000623. The minor difference between the two measurements falls within the margin of error.

Fig. R12. **a**, The RMCD curves sweep between +3 T and -3 T at 10 K, and the collection position is marked with a dashed circle in Fig. R12c. **b**, The RMCD signals shown along with the line sections of the RMCD image (c). **c**, RMCD maps obtained by using a 2.33 eV laser with diffraction-limited spatial resolution (see Methods), collected at 10 K and -0.75 T.

11) The author used the hBN flake as the insulating layer to prevent large leakage current. However, the *I-E* curves given in Fig. S4c indicated that the leakage current of hBN layer is about 1~4 nA. While for the hBN/NiI₂ device in Fig. 3b, the leakage current is only ~0.01nA. It implies that the NiI₂ trilayer is more resistive than the hBN layer. So I doubt the necessity of hBN inserting. Have the authors measure the *P-E* and *I-E* curves for the Gr/NiI₂/Gr devices?

Reply: We thank you for this helpful suggestion. We respectfully present a different viewpoint on the *I-E* curves. The observed current in the *I-E* curves (Supplementary Fig. 5c in the revised manuscript and Fig. 3b) is identified as displacement current rather than leakage current, as explained in Question 9. The hBN data in Fig. S4c (Supplementary Fig. 5c in revised manuscript) were recorded at room temperature, but the hBN/NiI₂ device in Fig. 3b was measured at 10 K. The huge temperature difference leads to the current change. Actually, the hBN layer is vital and necessary. The room-temperature *I-E* measurements have been conducted on the Gr/NiI₂/hBN/Gr device, revealing that the background current at 0.2 kHz (Fig. R13a) closely resembles that of the Gr/hBN/Gr device (Fig. R13b or Supplementary Fig. 5c). Moreover, Figure R13c shows the leakage current from the Gr/NiI₂/Gr and Gr/NiI₂/hBN/Gr devices at 10 K, demonstrating that the introduction of the insulating hBN layer effectively reduces the leakage current to near zero. But a serious leakage current takes place in the Gr/NiI₂/Gr device. Figure R13d illustrates the *P-E* behavior of the Gr/NiI₂/Gr device at 10 K and 1.3 Hz, displaying a ball-like loop characteristic of resistive leakages (*Ceram. Int.* **43**, 16676-16683, 2017; *J. Mater. Sci-Mater. El.* **29**, 19567-19577, 2018). Additionally, the *I-E* curve of the Gr/NiI₂/Gr device in Fig. R13e exhibits distinct linear current behavior rather than ferroelectric characteristics, indicating that the leakage current cancels the ferroelectric behavior of few-layer NiI₂. Thus, a high-quality insulating dielectric is crucial for assessing the ferroelectricity. The hBN flake serves as an effective insulating layer to mitigate substantial leakage current and enable the detection of ferroelectric features, as reported in previous studies (*Nat. Electron.* **4**, 98-108, 2021; *Nat. Electron.* **7**, 29-38, 2024).

Fig. R13. **a**, I - E characteristics measured at 0.2 KHz and 295 K for the trilayer NiI_2 device. **b**, I - E loop at 0.2 KHz and 295 K obtained from the $\text{Gr}/\text{hBN}/\text{Gr}$ device. **c**, Leakage-time characteristics measured at 10 K for the $\text{Gr}/\text{NiI}_2/\text{Gr}$ and $\text{Gr}/\text{NiI}_2/\text{hBN}/\text{Gr}$ devices. The insert shows the optical photograph of the $\text{Gr}/\text{NiI}_2/\text{Gr}$ device. **d**, **e**, P - V and I - V loops at 10 K from the $\text{Gr}/\text{NiI}_2/\text{Gr}$ device.

Reviewer #2 (Remarks to the Author):

Wu et al. investigated a few layer NiI_2 flakes, and demonstrated a coexistence of ferroelectricity and antiferroelectricity as well as a magnetic control of the ferroelectricity. Multiferroicity in 2D limit is an interesting and important issue, and authors have discussed it using an appropriate way. Followings are my comments about several issues to be clarified. Overall, authors have discussed important issues about the magnetoelectricity in a tri-layer multiferroic NiI_2 . I would be willing to reconsider this work if authors can improve it reflecting my comments.

Reply: We are grateful to you for your interest in and recognition of the significance of our manuscript. We also appreciate the time and effort they have dedicated to evaluating our work. In response to your comments, we have conducted theoretical calculations

on the magnetic configurations of trilayer NiI₂ and performed additional experiments to address the concerns of our respected reviewer.

1) Authors concluded the antiferromagnetic orders at 10 K from the MCD result showing zero remnant signal. This result can deny the ferromagnetic order, but does this already exclude any possibility to have a paramagnetic or diamagnetic state instead of the antiferromagnetic state?

Reply: We thank you for this helpful suggestion. It is indeed challenging that the antiferromagnetic (AFM) orders of trilayer NiI₂ are solely concluded from the MCD result with zero remnant magnetization. The AFM orders are deduced based on the following reasons:

(1) The RMCD signal shows near step-like changes with increasing magnetic field (Fig. 2d and Supplementary Fig. 3b), which sharply contrasts with the nearly linear magnetization of paramagnetic and diamagnetic materials. Particularly, diamagnetic state must show a negative magnetic susceptibility, and the RMCD signal should decrease with increasing magnetic field, which is inconsistent with our observations. Additionally, the disparity between the red and blue curves in the RMCD loop reveals hysteresis (Fig. 2d and Supplementary Fig. 3b), unlike the behavior expected in paramagnetic materials, where hysteresis is absent.

(2) To confirm the magnetic ground state of trilayer NiI₂, we performed density functional theory (DFT) calculations. The results suggest that the magnetic ground state of trilayer NiI₂ is a spiral spin configuration with an in-plane projection of the \mathbf{q} -vector along the [210] direction (depicted by the pink arrow in Fig. R1d). To provide a clearer view of the magnetic order, only the top Ni layer is illustrated, and the magnetic moments of the trilayer Ni are presented in Fig. R2. Importantly, trilayer NiI₂ exhibits interlayer AFM coupling due to the larger second interlayer nearest neighbor exchange parameter (Fig. R1e).

(3) Atomistic spin model simulations have been conducted and demonstrate that spiral spin states give rise to stripy domains (Fig. R1f), suggesting that the spiral spin state is the ground state of trilayer NiI₂, which is consistent with the Monte Carlo simulations

(*Phys. Rev. Lett.* **131**, 036701, 2023). To clearly show the magnetic structure, Fig. R1f displays only the spins of the initial Ni layer, while the spin textures of the other two layers are presented in Fig. R3a and c. The interlayer AFM coupling induces opposite magnetic moment components in adjacent layers.

(4) Variable-temperature micro-Raman spectroscopy was conducted in co-polarized channels ($\sigma+\sigma+$ channel) (Fig. R4a). At a temperature of 10 K, an observable peak at approximately 76 cm^{-1} was detected. A novel peak at around 30 cm^{-1} also appeared, associated with the magnon mode stemming from antiferromagnetic ordering. As the temperature rises, the peaks at 76 and 30 cm^{-1} progressively diminish, reaching a transition temperature of around 35 K (Fig. R4b). This finding is consistent with the magnetic phase transition temperature of AFM orders previously reported for trilayer NiI_2 (*ACS Nano* **14**, 10544-10551, 2020; *Nano Lett.* **21**, 5126-5132, 2021).

In summary, our comprehensive dataset—including DFT calculations, atomistic spin model simulations, RMCD, ferroelectric hysteresis loops, magnetic-field-induced ferroelectricity, and variable-temperature Raman spectroscopy—provides substantial evidence supporting the assertion that trilayer NiI_2 exhibits AFM behavior. The revised manuscript includes new data and representations.

Fig. R1. **a, b**, P - E and I - E loops at various frequencies from device 1 (D1). **c**, Corresponding I - E loops from Fig. R1b subtracted the current background. Two pairs of current peaks (FE-AFE and AFE-FE switching peaks) were obtained by Lorentz fitting. An evolution from FE to AFE was observed. **d**, Schematics of spiral configuration in top views. **e**, Schematics of spiral configuration in side views, showing the interlayer AFM coupling and out-of-plane ferroelectricity. **f**, Spin texture of NiI₂ from atomistic spin model simulations. Spins are represented by arrows, with the red and blue colors showing positive and negative values of the S_z component. **g**, The corresponding dipole texture obtained through calculations using the KNB model. Dipoles are represented by arrows, with the red and blue colors showing positive and negative values of the P_z component. **h**, Enlarged view of the out-of-plane AFE domain occurring in panel **g** (light blue solid line box).

Fig. R2. Top views of the magnetic order of trilayer NiI_2 . The blue, green, and gray balls represent Ni in the first, second, and third layers, respectively, while the arrows indicate the spin of Ni.

Fig. R3. **a, c**, Spin textures and corresponding dipole textures (**b, d**) from theoretical simulations. Spins and dipoles are represented by arrows, with red and blue colors indicating positive and negative values of the out-of-plane component. The scale bars are 2 nm.

Fig. R4. a, Circular polarization-resolved Raman spectra of 3L NiI₂ with temperature ranging from 10 K to 45 K. **b,** Temperature-dependent Raman peak intensities of the E_g and M modes (related to magnon scattering).

2) Authors stated that magnetic moments point upwards or downwards in the core region. What is the evidence of this argument?

Reply: We would like to express our gratitude to you for raising the question. The RMCD technique is equivalent to the polar magneto-optical Kerr effect (p-MOKE), which has been widely applied to image skyrmions (*Nature comm.* **13**, 722, 2022).

The RMCD imaging technique serves as a reliable and powerful method for evaluating two-dimensional magnetism at the microscale, and the RMCD intensity is proportional to the out-of-plane magnetization (*Nature* **546**, 270-273, 2017; *Nature* **604**, 468-473, 2022). The spatial distribution of RMCD signals can be correlated with the spatial distribution of out-of-plane magnetic moments (m_z). The presence of blue and red domains in paired formations is evident in both Fig. 2a and Supplementary Fig. 3. Figure 2b illustrates a specific pair of domains, while Fig. 2c shows the polar RMCD signals along the line sections of the RMCD map depicted in Fig. 2b. Within each paired domain, the RMCD signals exhibit opposite signs and nearly identical intensities (Fig. 2c). Notably, the RMCD signal is most pronounced in the core region, gradually diminishing as it extends towards the perimeter, eventually approaching zero (as

illustrated in Fig. 2c). This observation suggests that the spins within the core region are aligned parallel to the z-axis (either spin-up or spin-down), resulting in substantial magnetic moments. As the spins move away from the core, they progressively incline towards the in-plane orientation. In the vicinity of the boundary region, the magnetic moments lie entirely within the plane, leading to a near-zero magnetic moment.

3) What is the evidence of the non-collinear AFM order in the trilayer NiI₂?

Reply: This question is closely associated with Question 1. For a detailed explanation, please refer to the response to Question 1.

4) Can authors have any information about the q -vector in the magnetic ordered state?

This information is crucial in discussing the ferroelectricity based on Eq. (1).

Reply: We appreciate you for the helpful comments and suggestions. This question is closely associated with Question 1 and 3. The DFT calculations demonstrate that the magnetic ground state of trilayer NiI₂ exhibits a spiral spin configuration with an in-plane projection of the q -vector along the [210] direction (indicated by the pink arrow in Fig. R1d).

5) What is the length scale of Fig. 2b? And, on which area was the result of Fig. 2d obtained?

Reply: Fig. 2b presents an enlarged RMCD map of the region delineated by the white dashed box in Fig. 2a (middle panel), with a scale bar of 1 μm . The RMCD loop shown in Fig. 2d was collected in the region enclosed by the black circle in Fig. 2a (middle panel). We have modified Fig. 2 in the revised manuscript, along with the corresponding descriptions.

6) I-E and P-E curves seem clear. To strengthen the arguments of authors, there should be another measurement results of (bulk-like) thicker flakes which should exhibit typical behaviors of FE order with an absence of AFE order.

Reply: We are grateful to you for your helpful suggestions. A bulk NiI₂ device has been

prepared and measured, as illustrated in Fig. R5a. The evolution from ferroelectricity to antiferroelectricity with varying frequencies is also observed (Fig. R5b). In fact, theoretical studies have demonstrated that trilayer NiI₂, with its spiral magnetic orders, has already reached the bulk limit (*Phys. Rev. B.* **109**, 195422, 2024). Our atomistic spin model simulations and numerical calculations have revealed the coexistence of ferroelectric and antiferroelectric domains in each layer (Figs. R1g, h and Figs. R3b, d). This suggests that the coexistence of ferroelectric and antiferroelectric properties should also be manifested in the bulk material (Fig. R5). For a comprehensive discussion, please refer to the response section of Question 7.

Fig. R5. a, The optical photograph of the Gr/bulk NiI₂/hBN/Gr device. **b**, *P-E* loops at various frequencies obtained from the bulk NiI₂ device.

7) The arguments about the antiferroelectric state is not clear. Authors seem to explain simply double-well potential in the FE state and a possible spatial coexistence of negative and positive polarizations. As this explanation can be applied to the multi-domain state in the ferroelectric state, I would like to see an additional explanation for the antiferroelectric state.

Reply: We greatly appreciate you for the helpful comments and suggestions. Our

atomistic spin model simulations and computational calculations have revealed the coexistence of ferroelectric domains with opposite P_z components and interweaving local dipoles with opposing P_z components, and both the multi-domains and domain crossover probably contribute to the AFE behaviors. We have revised the explanations on antiferroelectric.

Atomistic spin model simulations have been conducted and demonstrate that spiral spin states give rise to stripy domains (Fig. R1f), suggesting that the spiral spin state is the ground state of trilayer NiI₂, which is consistent with the Monte Carlo simulations (*Phys. Rev. Lett.* **131**, 036701, 2023). In accordance with the KNB or inverse D-M mechanism, the dipole ordering configuration can be deduced from the spiral spin texture (Fig. R1g and Figs. R3b, d). The ferroelectric domains with opposite P_z components attributed to the distinct chirality of the magnetic domains (highlighted by the areas enclosed by the dashed red and black lines in Fig. R1g), because the spiral magnetic states with opposite chirality are degenerated but with opposite ferroelectric polarizations (*2D Mater.* **9**, 025010, 2022; *Phys. Rev. B* **109**, 195422, 2024). Remarkably, in certain transitional regions, local dipoles with opposing P_z components are interwoven, which induces antiferroelectricity in the out-of-plane direction (Fig. R1h). The calculated intricate intermingling of ferroelectric and antiferroelectric domains strongly supports the experimentally observed coexistence of ferroelectricity and antiferroelectricity. The interlayer AFM coupling induces opposite magnetic moment components in adjacent layers, but the spin chirality remains the same, which leads to the ferroelectric dipole textures remaining unaltered (Figs. R1f, g, and Fig. R3).

8) Authors observed 7% change of the polarization using a magnetic field, and they explained this observation based on a tilt of the spin rotation plane and the corresponding polarization flop. Does this process correspond to the polarization switching? Or, is there any possibility to have a gradual rotation of the spin rotation plane or ferroelectric polarization?

Reply: We appreciate you for the comments and suggestions. We agree that the magnetic control of ferroelectric polarization originates from a gradual rotation of the

spin rotation plane. The ferroelectric behavior in spiral magnets is elucidated by the inverse D-M interaction, expressed as $\mathbf{P} \propto \mathbf{e} \times \mathbf{q}$. In the top panel of Fig. R6, the spins rotate in the ac -plane ($\mathbf{e} // \mathbf{b}$), forming a transverse spiral with $\mathbf{q} // \mathbf{a}$, resulting in $\mathbf{P} // \mathbf{c}$. The middle panel illustrates that an applied magnetic field $\mathbf{H} // \mathbf{c}$ induces magnetization $\mathbf{M} // \mathbf{H} // \mathbf{c}$, leading to a transverse conical spin configuration with an effective ab rotation plane where $\mathbf{e} // \mathbf{H} // \mathbf{c}$ (*Phys. Rev. Lett.* **105**, 187601, 2010; Mostovoy, *Phys. Rev. Lett.* **96**, 067601, 2006). Consequently, a reorientation from $\mathbf{P} // \mathbf{c}$ to $\mathbf{P} // \mathbf{b}$ is anticipated, indicating a decreasing out-of-plane ferroelectric polarization. This scenario supports our experimental results that an out-of-plane magnetic field leads to out-of-plane magnetization (Fig. 2d) and a decrease of out-of-plane ferroelectric polarization (Figs. 4a-c). When $\mathbf{H} // \mathbf{q} // \mathbf{a}$, a longitudinal conical spin configuration with an effective bc rotation plane is established, as depicted in the bottom panel of Fig. R6. In this scenario ($\mathbf{e} // \mathbf{q}$), the ferroelectric polarization is expected to be zero according to the inverse D-M model ($\mathbf{P} \propto \mathbf{e} \times \mathbf{q}$). Overall, the magnetic field aligns the \mathbf{e} -vector with the magnetic field, leading to the reorganization of ferroelectric polarization.

To further validate the magnetic control of the \mathbf{e} -vector, **an in-plane magnetic field** has been applied to **manipulate the ferroelectric polarization** in configurations parallel and perpendicular to the \mathbf{S} -vector (sample orientation), as illustrated in Figs. R7a and c. Interestingly, for $\mathbf{H} // \mathbf{S}$, the out-of-plane ferroelectric polarization decreases with increasing magnetic field (Fig. R7b). In stark contrast, for $\mathbf{H} \perp \mathbf{S}$, the ferroelectric polarization increases with increasing magnetic field (Fig. R7d). In trilayer NiI_2 , the magnetic ground state adopts a spiral configuration with a propagation vector \mathbf{q} along the [210] direction (indicated by the pink arrow in Fig. R7e, f), while the \mathbf{e} -vector is perpendicular to the \mathbf{a} -axis, denoted by the green arrow. When the in-plane magnetic field is oriented between the \mathbf{e} -vector and the \mathbf{q} -vector, it steers the \mathbf{e} -vector to align parallel to the magnetic field direction, reducing the angle between the \mathbf{e} -vector and the \mathbf{q} -vector and consequently diminishing the ferroelectric polarization (Fig. R7g). Conversely, when the magnetic field is perpendicular to the \mathbf{S} -vector, it increases the angle between the \mathbf{e} -vector and the \mathbf{q} -vector, resulting in an enhancement of the

ferroelectric polarization (Fig. R7h). The experimental results are consistent with the aforementioned magnetoelectric coupling mechanism (Fig. R6).

Fig. R6. Schematics of ac -plane transverse spin spiral with propagation vector $q \parallel a$ (top panel), which induces ferroelectric polarization $P \parallel c$. Application of magnetic field ($H \parallel c$, middle panel) and ($H \parallel a$, bottom panel) is expected to stabilize the transverse and longitudinal spin conical with magnetization $M \parallel H$, respectively, where $P \parallel a$ ($P=0$) is expected within the KNB model.

Fig. R7. a, c, Optical photographs of the sample at different geometric configurations where the magnetic field is parallel or perpendicular to S -vector (sample orientation). The direction of the magnetic field is fixed, and the parallel ($H \parallel S$) and perpendicular configurations ($H \perp S$) are achieved by rotating the sample. b, c, P_r extracted from the

P - E hysteresis loop as a function of in-plane magnetic field at $\mathbf{H} // S$ and $\mathbf{H} \perp S$. \mathbf{e} , Schematics of spiral configuration in top views. \mathbf{f} , \mathbf{g} , \mathbf{h} , Magnetic-field control behavior of \mathbf{e} vs \mathbf{q} .

9) *Related to this issue, authors should provide a more discussion by comparing the RMCD and P_r obtained as a function of the magnetic field. In the RMCD loop, they observed a signature corresponding to spin flop process. According to the explanation given in the manuscript, I would expect a more dramatic variation of P_r at the corresponding magnetic field.*

Reply: We sincerely thank you for the helpful suggestions. In connection with question 8, an applied magnetic field $\mathbf{H} // c$ induces magnetizations $\mathbf{M} // \mathbf{H} // c$, leading to a transverse conical spin configuration with an effective ab rotation plane where $\mathbf{e} // \mathbf{H} // c$ (middle panel in Fig. R6). The reorientation from $\mathbf{P} // c$ to $\mathbf{P} // b$ is anticipated, indicating a decreasing out-of-plane ferroelectric polarization, which are consistent with that an out-of-plane magnetic field results in out-of-plane magnetization (Fig. 2d) and a decreasing out-of-plane ferroelectric polarization (Figs. 4a-c). **While the RMCD loop only indicates the out-of-plane magnetizations of a very local position, and the ferroelectric polarization P_r were net signals obtained in the whole trilayer NiI_2 ($\sim 188 \text{ um}^2$), thus it is hard to quantitatively link them together with magnetic field.** Hence, there may not be a directly correlation between the local RMCD and the net ferroelectric polarization P_r . Qualitatively, our in-plane and out-of-plane magnetic field control of ferroelectric polarization is consistent with the magnetoelectric coupling shown in Fig. R6, which is also supported by atomistic spin model simulations and computational calculations.

10) *It will be useful to compare the polarization controllability authors observed for NiI_2 with those of other multiferroics including CoI_2 . Also, there should be references cited for the statement "...~7%, ... which is remarkable feature of multiferroic." Actually, Ju et al. reported the change in the ferroelectric polarization for the 6L NiI_2 .*

(Nano Lett. 21, 5126 (2021)).

Reply: We appreciate you for the valuable suggestion. The literature that you recommended has been cited. A comprehensive review of previous studies on the magnetic field-induced polarization of bulk NiI₂, and few-layer NiI₂ is indeed necessary (*Phys. Rev. B* **87**, 014429, 2013; *Nano Lett.* **7**, 5126-5132, 2022). However, quantitatively comparing the magnitude of our magnetically-controlled ferroelectricity with the existing literature is challenging due to the differing orientations of ferroelectric polarization and applied magnetic field. All previous researches examined the magnetic control of in-plane ferroelectric polarization, but our work is the first to study the magnetic control of out-of-plane ferroelectric polarization. Moreover, the criteria are hard to be unified, and there is a huge difference, even for in-plane ferroelectric polarization, as shown in the following table.

	Configuration	Control ratio of Pr	Reference
Bulk NiI ₂	In-plane	~10% /T	Phys. Rev. B 87 , 014429, 2013
Few-layer NiI ₂	In-plane	~575% /T	Nano Lett. 7 , 5126-5132, 2022
Trilayer NiI ₂	Out-of-plane	~1% /T	This work

11) In the conclusion section, authors emphasized that NiI₂ “will introduce a paradigm shift for engineering new ultra-compact magnetoelectric devices.” I have to say this is over-stated. NiI₂ may show the ME coupling in tri-layer, but this phenomenon can be observed only at the very low temperature. I think this statement diminish many other efforts to realize the ME coupling at room temperature.

Reply: We extend our appreciation to you for the valuable suggestions. We have deleted this sentence.

12) There are several parts where English should be improved. Figures can be better described in the main text instead of repeating the text given in the figure caption. (Figure 4e, for example)

Reply: We have revised several expressions to enhance the accuracy of the language.

Reviewer #3 (Remarks to the Author):

Multiferroic materials displaying a strong magnetoelectric coupling are a promising platform for developing technological applications that require the control of magnetic orders using electric fields. Recently, evidence of multiferroic order has been reported in NiI₂ down to the few-layer [Hwiin Ju, et al Nano Letters 21, 5126–5132 (2021)] and monolayer limits [Qian Song, et al Nature 602, 601–605 (2022), Mohammad Amini, et al Advanced Materials, 2311342 (2024)]. These experiments have established NiI₂ as a new building block with huge potential for engineering van der Waals heterostructures with novel functionalities. However, proving multiferroicity in the monolayer limit by optical techniques is challenging [Jiang, Y. et al. Nature 619, E40-E43, (2023)] and new strategies need to be developed to demonstrate and characterize the multiferroic order in the low dimensional limit [Mohammad Amini, et al Advanced Materials, 2311342 (2024)]. Moreover, the evolution of the multiferroic order from the bulk to the single-layer limit remains unclear, and further studies need to be performed.

The authors of this work present an experimental analysis of a 3-layer NiI₂ device. Using scanning reflective magnetic circular dichroism microscopy and ferroelectric hysteresis loops, the authors show evidence of a non-collinear magnetic order and measure the out-of-plane component of the electric polarization in 3-layer NiI₂. Moreover, the authors show control of the ferroelectric polarization using an external magnetic field. From the experimental point of view, this study is timely and represents an important advance in the field of multiferroics and van der Waals materials. However, the theoretical interpretation of the experimental results shows some major inconsistencies that need to be amended before further consideration. If the authors address the following comments in the revised version of the manuscript, I will support the publication of this work in Nature Communications.

Reply: We greatly appreciate you for the helpful suggestions and support of our manuscript for publication. We deeply appreciate your expertise and professional acumen in the NiI₂ system, along with your insightful grasp of the key issues in the realm of two-dimensional multiferroics. Based on your constructive suggestions, we

have done new experimental and theoretical studies on trilayer NiI₂, which have been added into Fig. 3 and Supplementary information in the revised manuscript.

(1) DFT calculations demonstrates that the magnetic ground state of trilayer NiI₂ is a spiral spin configuration with an in-plane projection of the \mathbf{q} -vector along the [210] direction (depicted by the pink arrow in Fig. R1d). The magnetic moments of the trilayer Ni are presented in Fig. R2. Importantly, trilayer NiI₂ exhibits interlayer AFM coupling due to the larger second interlayer nearest-neighbor exchange parameter (Fig. R1e). And an in-plane projection of the \mathbf{e} -vector is perpendicular to the direction of \mathbf{a} (indicated by the green arrow in Fig. R1d), resulting in an out-of-plane components (c -axis direction) of ferroelectric polarization (Fig. R1d and e), which support the experimentally observed out-of-plane ferroelectricity (Fig. R1a).

(2) Atomistic spin model simulations demonstrate that spiral spin states give rise to stripy domains (Fig. R1f, R3a and R3c), which further validate that the spiral spin state is the magnetic ground state of trilayer NiI₂, consistent with the stripe domains predicted by Monte Carlo simulations (*Phys. Rev. Lett.* **131**, 036701, 2023).

(3) Computational simulations have revealed the coexistence of ferroelectric domains with opposite \mathbf{P}_z components (Fig. R1g and Figs. R3b, d), which are attributed to the distinct chirality of the magnetic domains (highlighted by the areas enclosed by the dashed red and black lines in Fig. R1g), because the spiral magnetic states with opposite chirality are degenerated but with opposite ferroelectric polarizations (*2D Mater.* **9**, 025010, 2022; *Phys. Rev. B* **109**, 195422, 2024). Notably, in certain transitional regions, local dipoles with opposing \mathbf{P}_z components are interwoven, which probably give rise to the antiferroelectricity in the out-of-plane direction (Fig. R1h). The theoretically intricate intermingling of ferroelectric and antiferroelectric domains supports the experimentally observed coexistence of ferroelectricity and antiferroelectricity. The spatial distributions of the \mathbf{S}_z components in adjacent layers exhibit opposite characteristics, yet their dipole textures remain predominantly unaltered (Figs. R1f, g, and Fig. R3). This phenomenon is attributed to the interlayer antiferromagnetic coupling (Fig. R1e), which results in opposite signs of the \mathbf{S}_z components in adjacent layers while maintaining the chirality of the spiral magnetic order within each layer.

Fig. R1. **a, b,** P - E and I - E loops at various frequencies from device 1 (D1). **c,** Corresponding I - E loops from Fig. R1b subtracted the current background. Two pairs of current peaks (FE-AFE and AFE-FE switching peaks) were obtained by Lorentz fitting. An evolution from FE to AFE was observed. **d,** Schematics of spiral configuration in top views. **e,** Schematics of spiral configuration in side views, showing the interlayer AFM coupling and out-of-plane ferroelectricity. **f,** The spin texture of NiI_2 from atomistic spin model simulations. Spins are represented by arrows, with the red and blue colors showing positive and negative values of the S_z component. **g,** The corresponding dipole texture obtained through calculations using the KNB model. Dipoles are represented by arrows, with the red and blue colors showing positive and negative values of the P_z component. **h,** An enlarged view of the out-of-plane AFE

domain occurring in panel **g** (light blue solid line box).

Fig. R2. Top views of the magnetic order of trilayer NiI_2 . The blue, green, and gray balls represent Ni in the first, second, and third layers, respectively, while the arrows indicate the spin of Ni.

Fig. R3. **a, c**, Spin textures and corresponding dipole textures (**b, d**) from theoretical simulations. Spins and dipoles are represented by arrows, with red and blue colors indicating positive and negative values of the out-of-plane component. The scale bars are 2 nm.

1) The authors identify the number of NiI₂ layers in their system using the Raman feature associated with the interlayer shear mode as explained in reference 20 of the manuscript. Could the authors also provide the optical contrast for the identification of the number of layers as also detailed in reference 20?

Reply: We extend our appreciation to you for your valuable suggestions. To measure the ferroelectric hysteresis loops, the NiI₂ flake has to be placed on the bottom graphene electrode with the hBN insulator layer, and further covered with the top graphene electrode. Moreover, NiI₂ is extremely air-sensitive and needs to be encapsulated with another hBN flake for stabilization. The complex multi-layered stacking structure of the device poses challenges for accurately identifying the thickness of NiI₂ using the optical contrast analysis.

To further verify the layer thickness, variable-temperature micro-Raman spectroscopy has been utilized in co-polarized channels ($\sigma^+\sigma^+$ channel), as detailed in Fig. R4a. At 10 K, two peaks at approximately 76 and 30 cm^{-1} were observed, which are assigned to the E_g and magnon mode, respectively. With increasing temperature, the peaks at around 76 and 30 cm^{-1} gradually diminish, revealing a transition temperature of approximately 35 K (Fig. R4b), which is well consistent with the previously reported magnetic transition temperature for trilayer NiI₂ (*ACS Nano* **14**, 10544-10551, 2020; *Nano Lett.* **21**, 5126-5132, 2021; *arXiv:2307.10686*, 2023). Thus, NiI₂ is considered as trilayer through its interlayer shear mode and magnetic transition temperature. The new data and representations have been added in the revised manuscript.

Fig. R4. a, Circular polarization-resolved Raman spectra of 3L NiI₂ with temperature ranging from 10 K to 45 K. **b,** Temperature-dependent Raman peak intensities of the E_g and M modes (related to magnon scattering).

2) In caption (a) of Fig 2, the authors state that the RMCD maps are taken at room temperature, in the plot Fig 2a they show T=10K. I suppose that the latter is the right temperature of the maps, the authors should correct the caption.

Reply: Thanks a lot for this suggestion. We have revised the caption accordingly.

3) Regarding the RMCD maps, Could the authors present in the SI more maps at different magnetic field values to show that the evolution with the magnetic field that they detail is robust?

Reply: We appreciate you for the suggestions. More RMCD maps (Figs. R5c-j) at different magnetic field intensity have been added to the Supplementary information (Supplementary Fig. 3).

Fig. R5 a, Optical micrograph of another few-layer NiI₂ put on hBN substrate and further covered by another hBN on the top. The white and red dashed-line box represents the profile of the NiI₂ flake and the area of RMCD maps. **b**, The RMCD curves sweeping between +3 T and -3 T at 10 K, suggesting a non-collinear antiferromagnetism. The result was collected from the area enclosed by the black circle in **j**. **c-j**, Polar RMCD maps at 10 K, taken at selected out-of-plane magnetic field.

4) The RMCD as a function of the magnetic field is shown in Fig. 2d allowing us to identify a non-collinear magnetic order. It would be highly interesting if the authors could present results from 7T to 7T as they do for the electric polarization in Fig 4. This would help to clarify their interpretation of the decrease in polarization when applying an external magnetic field. One would think that if the spin spiral is aligned in the xy plane when an external magnetic field is applied parallel to the z direction, the RMCD signal should decrease since the RMCD intensity is proportional to the out-of-plane magnetization, which will go to zero if the spiral is aligned on the xy plane. However, if a transverse conical spin spiral is also formed when applying an external magnetic field, the RMCD signal would increase while decreasing the out-of-plane electric polarization (see comments 7 and 8 for more details about the transverse conical spin spiral). The author should review this point and if possible show measurements of the RMCD signal from -7T to 7T as they do for the electric polarization.

Reply: We appreciate you for your professional suggestions. We apologize that our previous writing and explanations were unclear and may have misled readers. We have added more detailed explanations in the revised manuscript. We agree with you that a transverse conical spin spiral is formed upon the application of an external magnetic field. And we are very sorry for that our system has limitation to do the micro-optical measurements under 7 T due to the vibrations from GM cryocooler of the superconducting magnet ([arXiv:2307.05363](https://arxiv.org/abs/2307.05363)).

In the revised manuscript, new experiments and theoretical calculations, including DFT simulations, atomistic spin model simulations and numerical simulations, as well as the in-plane magnetic control experiments of ferroelectric polarization, have been added to deeply look into the magnetoelectric coupling mechanism. The ferroelectric behavior in spiral magnets is elucidated by the inverse D-M interaction, expressed as $\mathbf{P} \propto \mathbf{e} \times \mathbf{q}$, where \mathbf{q} represents the spiral's propagation vector and \mathbf{e} denotes the rotational axis of the spin-rotation-plane. In the top panel of Fig. R6, the spins rotate in the ac -plane ($\mathbf{e} \parallel \mathbf{b}$), forming a transverse spiral with $\mathbf{q} \parallel \mathbf{a}$, resulting in $\mathbf{P} \parallel \mathbf{c}$. The middle panel illustrates that an applied magnetic field $\mathbf{H} \parallel \mathbf{c}$ induces magnetization $\mathbf{M} \parallel \mathbf{H} \parallel \mathbf{c}$, leading to a transverse conical spin configuration with an effective ab rotation plane where $\mathbf{e} \parallel \mathbf{H} \parallel \mathbf{c}$ (*Phys. Rev. Lett.* **105**, 187601, 2010; *Phys. Rev. Lett.* **96**, 067601, 2006). Consequently, a reorientation from $\mathbf{P} \parallel \mathbf{c}$ to $\mathbf{P} \parallel \mathbf{b}$ is anticipated, indicating a decreasing out-of-plane ferroelectric polarization. This scenario supports our experimental results that an out-of-plane magnetic field leads to out-of-plane magnetization (Fig. 2d) and a decrease of out-of-plane ferroelectric polarization (Figs. 4a-c). When $\mathbf{H} \parallel \mathbf{q} \parallel \mathbf{a}$, a longitudinal conical spin configuration with an effective bc rotation plane is established, as depicted in the bottom panel of Fig. R6. In this scenario ($\mathbf{e} \parallel \mathbf{q}$), the ferroelectric polarization is expected to be zero according to the inverse D-M model ($\mathbf{P} \propto \mathbf{e} \times \mathbf{q}$). Overall, the magnetic field aligns the \mathbf{e} -vector with the magnetic field, leading to the reorganization of ferroelectric polarization. To further validate the magnetic control of the \mathbf{e} -vector, an in-plane magnetic field has been applied to manipulate the ferroelectric polarization in configurations parallel and perpendicular to the \mathbf{S} -vector (sample orientation), as

illustrated in Fig. R7a and c. Interestingly, for $\mathbf{H} \parallel \mathbf{S}$, the out-of-plane ferroelectric polarization decreases with increasing magnetic field (Fig. R7b). In stark contrast, for $\mathbf{H} \perp \mathbf{S}$, the ferroelectric polarization increases with increasing magnetic field (Fig. R7d). In trilayer NiI_2 , the magnetic ground state adopts a spiral configuration with a propagation vector \mathbf{q} along the $[210]$ direction (indicated by the pink arrow in Fig. R7e, f), while the \mathbf{e} -vector is perpendicular to the \mathbf{a} -axis, denoted by the green arrow. When the in-plane magnetic field is oriented between the \mathbf{e} -vector and the \mathbf{q} -vector, it steers the \mathbf{e} -vector to align parallel to the magnetic field direction, reducing the angle between the \mathbf{e} -vector and the \mathbf{q} -vector and consequently diminishing the ferroelectric polarization (Fig. R7g). Conversely, when the magnetic field is perpendicular to the \mathbf{S} -vector, it increases the angle between the \mathbf{e} -vector and the \mathbf{q} -vector, resulting in an enhancement of the ferroelectric polarization (Fig. R7h). The experimental results are consistent with the aforementioned magnetoelectric coupling mechanism (Fig. R6).

Fig. R6. Schematics of ac -plane transverse spin spiral with propagation vector $\mathbf{q} \parallel \mathbf{a}$ (top panel), which induces ferroelectric polarization $\mathbf{P} \parallel \mathbf{c}$. (Application of magnetic field ($\mathbf{H} \parallel \mathbf{c}$, middle panel) and [$\mathbf{H} \parallel \mathbf{a}$, bottom panel] is expected to stabilize the transverse and longitudinal spin conical with magnetization $\mathbf{M} \parallel \mathbf{H}$, respectively, where $\mathbf{P} \parallel \mathbf{a}$ [$\mathbf{P}=0$] is expected within the KNB model.

Fig. R7. **a, c**, Optical photographs of the sample at different geometric configurations where the magnetic field is parallel or perpendicular to S -vector (sample orientation). The direction of the magnetic field is fixed, and the parallel ($H // S$) and perpendicular configurations ($H \perp S$) are achieved by rotating the sample. **b, d**, P_r extracted from the P - E hysteresis loop as a function of in-plane magnetic field at $H // S$ and $H \perp S$. **e**, Schematics of spiral configuration in top views. **f, g, h**, Magnetic-field control behavior of e vs q .

5) In reference 42 the in-plane projection of the bulk q -vector along the $[210]$ (ie along the second-neighbor Ni atom direction) is theoretically considered to analyze the origin of the multiferroic order in NiI₂. However, it has been theoretically predicted (Sodequist et al 2D Materials 10, 035016 (2023)) and experimentally measured (Mohammad Amini, et al Advanced Materials, 2311342 (2024)) that the q in the monolayer goes along a different direction (third-neighbor Ni atom direction). For the 3-layer NiI₂ case that the authors present here, there is no evidence for the spin spiral q vector. It could be the q -vector shown in ref 42 if the bulk limit is already achieved in the 3-layer system (as suggested in ref 20), or a q -vector similar to the one found in the monolayer could arise.

Since this limit is not explored yet, I would suggest the authors warn the reader about this point.

Reply: We appreciate you for the professional suggestions. The trilayer NiI₂ has already reached the bulk limit (arXiv:2307.10686, 2023). The DFT calculations have revealed that the magnetic ground state of trilayer NiI₂ is a spiral spin configuration and an in-plane projection of the \mathbf{q} -vector is along the [210] direction (depicted by the pink arrow in Fig. R1d), which closely resembles that of the bulk system (*Phys. Rev. B.* **109**, 195422, 2024). The spin-polarized scanning tunneling microscopy (SPSTM) shows that the \mathbf{q} -vector in monolayer NiI₂ is determined as (0.2203, 0, 0), differing from that in trilayer (arXiv:2309.16526).

The theoretical interpretation that the authors provide for the decrease of electric polarization as a function of the external magnetic field presents inconsistencies.

*6) First, the picture that they sketch from their results states that in the absence of a magnetic field, the spin spiral develops in the xz rotation plane with an \mathbf{e} vector along the y direction producing an electric polarization in the z-direction. It has been theoretically shown that Kitaev interaction promotes a diagonal plane around 40 degrees for the spin spiral (Xuanyi Li, et al *Phys. Rev. Lett.* **131**, 036701 (2023)). It would be more consistent with previous analyses and experimental observations to describe the magnetic state in the absence of an external magnetic field as a spin spiral in an oblique plane, rather than in the limiting xz plane.*

7) Second, the authors propose that the external magnetic field modifies the plane of the spin spiral from xz to xy, which causes a reduction of the polarization along the z direction as shown by their results. However, this scenario should also cause a reduction of the RMCD signal as a function of the magnetic field, something at odds with the results presented in Fig 2d. The authors need to reconcile these two observations. The current theoretical interpretation is not consistent.

8) An alternative interpretation that I suggest the authors explore is the formation of a transverse conical spin spiral besides the reorientation of the spin spiral plane in the xy direction when applying the external magnetic field along the z-direction. This would

decrease of the electric polarization in the z-direction while simultaneously increasing the magnetization along the z-direction as a function of the magnetic field in the z-direction. In this regard, I suggest the authors review (also improve the quality of the sketches) of Fig 7 in the SI as well as Fig 3 d,e,f to make the evolution of the magnetic order with the magnetic field much clearer and consistent with all the experimental observations.

Reply: We greatly appreciate you for the professional comments and suggestions, which indeed helped us improve our work and manuscript. We agree with you that the magnetic control of ferroelectric polarization is induced through a transverse conical spin spiral when applying an external magnetic field.

As illustrated in Fig. R6, the spins exhibit rotation in the ac -plane ($\mathbf{e} // \mathbf{b}$), forming a transverse spiral with $\mathbf{q} // \mathbf{a}$, thereby achieving $\mathbf{P} // \mathbf{c}$ (top panel). When a magnetic field ($\mathbf{H} // \mathbf{c}$) is introduced, the magnetization $\mathbf{M} // \mathbf{H} // \mathbf{c}$ is induced, leading to a transverse conical spin configuration with an effective ab rotation plane where $\mathbf{e} // \mathbf{H} // \mathbf{c}$ (Mochizuki, M. et al. Theory of magnetic switching of ferroelectricity in spiral magnets. *Phys. Rev. Lett.* **105**, 187601, 2010; Mostovoy, M. et al. Ferroelectricity in spiral magnets. *Phys. Rev. Lett.* **96**, 067601, 2006). The reorientation of \mathbf{P} from $\mathbf{P} // \mathbf{c}$ to $\mathbf{P} // \mathbf{b}$ occurs, which in agreement with the experimental observations that an out-of-plane magnetic field in trilayer NiI_2 leads to an increase of out-of-plane magnetization (Fig. 2d) and a decrease of out-of-plane ferroelectric polarization (Figs. 4a-c).

But when $\mathbf{H} // \mathbf{q} // \mathbf{a}$, a longitudinal conical spin configuration with an effective bc rotation plane is established (bottom panel in Fig. R6). In this scenario ($\mathbf{e} // \mathbf{q}$), the ferroelectric polarization is predicted to be zero according to the inverse D-M model ($\mathbf{P} \propto \mathbf{e} \times \mathbf{q}$). Overall, the magnetic field aligns the \mathbf{e} -vector with the magnetic field, causing the rearrangement of ferroelectric polarization.

To further study the magnetoelectric coupling mechanism, in-plane magnetic fields have been applied to manipulate the ferroelectric polarization, which employs the parallel and perpendicular to the \mathbf{S} -vector (sample orientation), as illustrated in Figs. R7a and c. Interestingly, for $\mathbf{H} // \mathbf{S}$, the out-of-plane ferroelectric polarization decreases

with the magnetic field increase (Fig. R7b). In stark contrast, for $\mathbf{H} \perp \mathbf{S}$, the ferroelectric polarization increases as increasing magnetic field (Fig. R7d).

The DFT calculations demonstrate that the magnetic ground state of trilayer NiI_2 exhibits a spiral spin configuration with an in-plane projection of the \mathbf{q} -vector along the [210] direction (as indicated by the pink arrow in Fig. R7e, f). An in-plane projection of the \mathbf{e} -vector is perpendicular to the direction of \mathbf{a} (as indicated by the green arrow in Fig. R7e). When the in-plane magnetic field is oriented at an azimuth angle between the \mathbf{e} -vector and the \mathbf{q} -vector, it drives the \mathbf{e} -vector to align with the magnetic field direction, reducing the angle between the \mathbf{e} -vector and the \mathbf{q} -vector, thereby diminishing the ferroelectric polarization (Fig. R7g). Conversely, when the magnetic field is perpendicular to the \mathbf{S} -vector, it leads to an increase in the angle between the \mathbf{e} -vector and the \mathbf{q} -vector, resulting in an enhancement of the ferroelectric polarization (Fig. R7h). The experimental results are well consistent with our theoretical calculations. We express our gratitude to you again.

9) The references need to be highly improved. Relevant works in the field are missing: Hwiin Ju, et al Nano Letters 21, 5126–5132 (2021) shows evidence of non-collinear magnetic order down to the bilayer limit using Second Harmonic Generation, Mohammad Amini, et al Advanced Materials, 2311342 (2024) demonstrates and characterizes the multiferroic order of monolayer NiI_2 using Scanning Tunneling Microscopy, Xuanyi Li, et al Phys. Rev. Lett. 131, 036701 (2023) introduces a realistic spin model to understand the magnetic properties of NiI_2 , in particular the out-of-plane tilt of the spin spiral, Sodequist et al 2D Materials 10, 035016 (2023) theoretically shows that in the monolayer the \mathbf{q} vector of the spin spiral differs from the one in bulk. Some older, but also important references providing the first studies of the magnetic order in NiI_2 using neutron scattering: D. Billerey, et al Physics Letters A 61, 138–140 (1977) and S.R. Kuindersma, et al Physica B+C 111, 231–248 (1981).

Reply: We have added additional sentences and references to address these ideas.

10) *There is a typo in the summary paragraph, "hysteresis loop" is repeated.*

Reply: We express our gratitude to you again. We have deleted the repeated terms.

RESPONSE TO REVIEWERS' COMMENTS

Reviewer #1 (Remarks to the Author):

In the revised manuscript, the authors have done a lot of additional experimental works (such as SHE, LD and temperature dependent Raman spectra) and performed the DFT calculations as well as the atomistic spin model simulations. The responses have satisfactorily addressed most of my previous comments. I just have few minor comments.

Reply: We sincerely thank you for your considerable time and effort you dedicated for evaluating our work. Thank you for the positive feedback on the revised version. Your suggestions have significantly improved the quality of our manuscript.

1. The author mentioned that the observed background current in Fig. 4e should be identified as displacement current rather than conduction current. This is an important information. I suggest the author to include related discussion in the main text or Supplementary Materials. Also, maybe the authors can give the DC measurement of the resistance of the devices (including Gr/NiI₂/Gr, Gr/hBN/Gr and Gr/NiI₂/hBN/Gr) to prove this conclusion.

Reply: Thank you for your valuable suggestions. The discussion regarding displacement current has been added to the supplementary materials (Supplementary Note 3; highlighted in red). The device's resistance can be obtained through DC leakage current testing (Supplementary Fig. 6a), and the resistance value is shown in Supplementary Note 2 (highlighted in red).

2. The P-V and I-V loops from the Gr/NiI₂/Gr device give solid evidence that the introduction of the insulating hBN layer can effectively reduce the leakage current. I suggest the authors to include these figures in the Supplementary Materials.

Reply: We express our gratitude to you for the valuable suggestions. The data you recommended has been added to the supplementary materials (Supplementary Fig. 6).

3. I think the discussion on the Raman scattering intensity based on the Raman selection rule will be helpful to the readers who are not very familiar with Raman measurements. I suggest the authors to include the discussion in Supplementary Fig. 2.

Reply: Thank you for your helpful suggestions. The discussion regarding the Raman selection rules has been added to the supplementary materials (Supplementary Note 1; highlighted in red).

Lastly, we would like to express our sincere gratitude once again for your constructive feedback. Your suggestions have greatly improved both the scientific rigor and the writing quality of our manuscript.

Reviewer #2 (Remarks to the Author):

Authors replied comments and revised the manuscript satisfactorily. I have one thing to ask the authors to implement in the manuscript. Although the DFT calculations could

give information about the q -vector, it was not experimentally verified. Therefore, authors should be more careful when they discuss the magnetoelectricity based on the given q -vector. Authors may explicitly state that the q -vector is not experimentally verified.

Overall, this work clearly demonstrates the magnetoelectricity of the tri-layer NiI_2 which is worthy to be published in Nature Communications. I strongly recommend its publication.

Reply: We thank you for your recommendation. Your comments have been crucial in improving the overall quality of our work, and we sincerely appreciate your support for the publication. At present, the q -vector is confirmed by theoretical calculations. New technologies should be developed to detect the q -vector; however, this remains challenging.

Reviewer #3 (Remarks to the Author):

The authors have addressed all the comments raised in the previous report and modified the article accordingly. The inconsistencies observed in the previous version have been successfully resolved and the new version of the manuscript has been substantially improved. The work is timely and represents an important advancement in the fields of van der Waals materials and multiferroics. Therefore, I support the publication of the current version of this article in Nature Communications.

Reply: We sincerely appreciate your support in publishing our manuscript in *Nature Communications*.

RESPONSE TO REVIEWERS' COMMENTS

Reviewer #1 (Remarks to the Author):

The authors have modified the manuscript accordingly. I now believe publication in Nature Communications is warranted.

Reply: We sincerely appreciate your support in the publication of our manuscript in Nature Communications.